# NeSyGeo: A Neuro-Symbolic Framework for Multimodal Geometric Reasoning Data Generation

## Abstract

Obtaining large-scale, high-quality reasoning data is crucial for improving the geometric reasoning capabilities of multi-modal large language models (MLLMs). However, existing data generation methods, whether based on predefined templates or constrained symbolic provers, inevitably face diversity and numerical generalization limitations. To address these limitations, we propose NeSyGeo, a novel neuro-symbolic framework for generating geometric reasoning data. First, we propose a domain-specific language grounded in the entity–attributes-relations paradigm to comprehensively represent all components of plane geometry, along with generative actions defined within this symbolic space. We then design a symbolic–visual–text pipeline that synthesizes symbolic sequences, maps them to visual and textual representations and generates reasoning path with reverse search and forward validation. Based on this framework, we construct NeSyGeo-CoT and NeSyGeo-Caption datasets, containing 100k samples, and release a new benchmark NeSyGeo-Test for evaluating geometric reasoning abilities in MLLMs. Experiments demonstrate that the proposal significantly and consistently improves the performance of multiple MLLMs under both reinforcement and supervised fine-tuning. With only 4k samples and two epochs of reinforcement fine-tuning, base models achieve improvements of up to +15.8% on MathVision, +8.4% on MathVerse, and +7.3% on GeoQA. Notably, a 4B model can be improved to outperform an 8B model from the same series on geometric reasoning tasks.

## 1 Introduction

Improving the visual reasoning capabilities of MLLMs has garnered significant attention recently (Liu et al., 2023; Alayrac et al., 2022; Achiam et al., 2023; Li et al., 2021; Jiang et al., 2024; Zhang et al., 2025a; Wang et al., 2024a; Wu et al., 2024; Liang et al., 2024), with models like InternVL (Chen et al., 2024) and the QwenVL series (Wang et al., 2024c; Bai et al., 2025) demonstrating significant enhancements in visual-semantic comprehension through their multimodal capabilities. Among various visual reasoning tasks, geometric mathematical reasoning is crucial for evaluating the reasoning performance of MLLMs (Yan et al., 2025; 2024), as it requires a deep integration of spatial perception, symbolic understanding, and logical deduction. To enhance such reasoning abilities, existing approaches (Zhang et al., 2024d; 2025b; Zhao et al., 2025) primarily rely on fine-tuning base models using reinforcement learning (RL) or supervised fine-tuning (SFT) on specialized geometric reasoning datasets. These methods depend heavily on the availability of large-scale, high-quality geometric reasoning data, which is often costly and time-consuming to construct manually. Therefore, automatic data generation for geometric reasoning has emerged as a promising and actively explored direction, aiming to alleviate data scarcity and further improve the reasoning abilities of MLLMs.

Existing approaches for generating datasets in geometric tasks can be broadly classified into four categories. **Text augmentation methods** like G-LLaVA (Gao et al., 2025) primarily mutate the conditions of existing datasets through equivalent condition transformation and numerical scaling. However, this approach fails to address the scalability of image generation. **Template-based methods** (Deng et al., 2024; Zhang et al., 2024d; Kazemi et al., 2024), use predefined geometric templates with fixed topologies, simplifying synthesis but constraining diversity by reducing the geometric space to limited combinations. **Solver-based methods** (Huang et al., 2025; Zhang et al., 2024a) inspired by

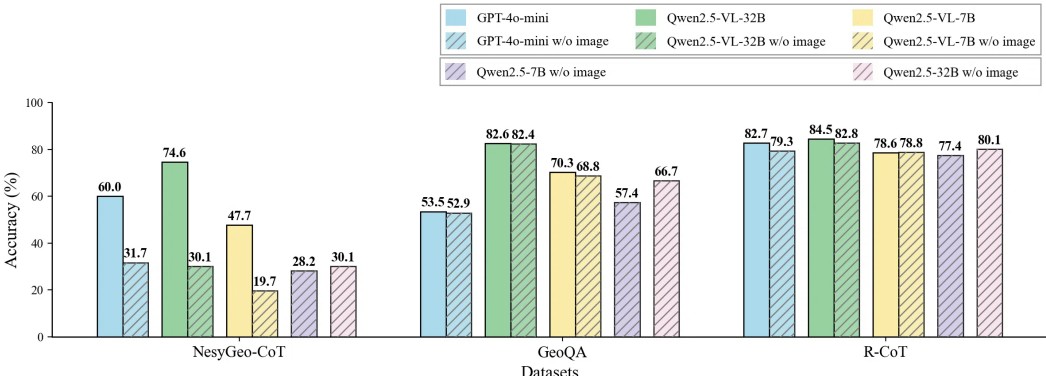

Figure 1: Performance comparison of different MLLMs and LLMs with and without image input in several geometry datasets. The minimal or negligible drops observed upon image removal in GeoQA and R-CoT raise concerns regarding the utilization of visual information for geometric reasoning.

symbolic prover AlphaGeometry (Trinh et al., 2024), leverage formal languages for synthesis but lack metric details (e.g., angles, lengths, areas), restricting multimodal data to descriptive annotations and limiting numerical reasoning applications.In summary, existing methods grapple with issues of image scalability, limited geometric diversity, a lack of precise numerical information, and challenges in ensuring the reliability of generated content.

Moreover, current geometric reasoning datasets present three critical limitations in cross-modal reasoning:1) **Information Redundancy**: Repetitive textual conditions provide a "shortcuts" that allows models to bypass visual reasoning. 2) **Semantic Gap Between Modalities**: Abstract geometric properties stated in the text (such as equality) are not explicitly grounded from the visual features. 3) **Poor Image Quality:** Low-resolution and inadequate images hinder the extraction of crucial visual information. As shown in Figure 1, our comparative experiments demonstrate minimal or negligible accuracy drops upon image removal. Such discrepancies hinder the model's ability to make use of visual perception and learn robust cross-modal reasoning .

To address these challenges, we propose **NeSyGeo**, a neuro-symbolic framework for synthesising high-quality multimodal geometric reasoning datasets. NeSyGeo integrates three components: 1) A formal geometric symbolic space defined by a domain-specific language (DSL), capturing primitive entities (points, lines, circles), metric attributes (angles, lengths) and topological relations (parallelism, incidence, perpendicularity), enabling diverse geometric configurations via systematic sampling within constrained parametric bounds. 2) Bidirectional conversion engines that transform symbolic constructs into decoupled modalities, producing annotated vector graphics paired with concise textual axioms. 3) Validated Question-Answer (Q&A) pairs and theorem-grounded CoT generators that effectively merges neural reasoning with verification.

Our framework enhances data quality by generating a wide spectrum of well-grounded images and correspondingly complex textual descriptions, while ensuring the validity and fidelity of both modalities. Furthermore, by carefully annotating semantic relationships within the image and avoiding redundancy between textual and visual conditions, it compels MLLMs to actively engage with visual information, thereby preventing modality neglect and significantly improving their visual grounding capabilities during training.

Leveraging the NeSyGeo pipeline, we construct two training datasets, NeSyGeo-Caption and NeSyGeo-CoT, comprising 100k samples. NeSyGeo-Caption aims to improve the perceptual understanding of geometric elements, while NeSyGeo-CoT primarily focuses on enhancing logical reasoning. The key characteristics of our dataset compared to other popular multimodal geometric datasets are presented in Figure 2. Additionally, we develop an evaluation set, NeSyGeo-Test, with 2668 Q&A pairs, enabling a thorough assessment of the geometric reasoning capabilities of mainstream MLLMs. Notably, our training dataset consistently and efficiently enhances the geometric reasoning performance of MLLMs across multiple benchmarks. With only 4k samples and two epochs of RL training, base models achieve performance improvements of up to **+15.8%** on MathVision, **+8.4**% on MathVerse, and **+7.3%** on GeoQA. Moreover, InternVL2.5-4B can be improved to outperform the 8B model in the same series on geometric reasoning tasks.

| Features → Datasets ↓ | Image Features | | | | | Text Features | | | | Total Features | | |
|---|---|---|---|---|---|---|---|---|---|---|---|---|
| | Number of Images | Automatic Synthesis | High Resolution | Visual Annotation | Symbolic Form | Number of QA Pairs | Caption Data | Reasoning Data | Step-by-step CoT | Classification of Difficulty | Classification of Elements | Visual Understanding |
| Geometry-3k | 2.1k | ✗ | ✗ | ✗ | ✔ | 2.1k | ✗ | ✗ | ✗ | ✗ | ✔ | ✗ |
| GeoQA | 3.5k | ✗ | ✗ | ✗ | ✗ | 5k | ✗ | ✔ | ✗ | ✗ | ✔ | ✗ |
| G-LLaVA | 8.1k | ✗ | ✗ | ✗ | ✗ | 110k | ✔ | ✔ | ✗ | ✗ | ✗ | ✗ |
| AutoGeo | 100k | ✔ | ✔ | ✗ | ✗ | 100k | ✔ | ✗ | ✗ | ✗ | ✔ | ✗ |
| GeomVerse | 10k | ✔ | ✔ | ✗ | ✗ | 10k | ✗ | ✔ | ✗ | ✔ | ✗ | ✗ |
| R-CoT | 33k | ✔ | ✔ | ✔ | ✗ | 87k | ✗ | ✔ | ✔ | ✔ | ✗ | ✗ |
| NeSyGeo | 85.3k | ✔ | ✔ | ✔ | ✔ | 100k | ✔ | ✔ | ✔ | ✔ | ✔ | ✔ |

Figure 2: Comparison of dataset characteristics synthesized by our method and other popular synthesis approaches. "High Resolution" denotes average image pixels exceeding 336×336. "Symbolic Form" refers to the symbolic meta-information associated with the image. "Classification of Elements" signifies categorization by geometric elements. "Visual Understanding" represents the mitigation of image-text redundancy for stronger visual grounding in reasoning. More specific examples of different methods are in Appendix E.1.

In summary, our contributions are as follows:

- We propose **NeSyGeo**, a novel neuro-symbolic framework for geometric reasoning data generation. NeSyGeo incorporates a comprehensive Geo-DSL, a reasoner–verifier for CoT generation, and a painter–translator module for informalization. It ensures data quality, enhances diversity, improve the correctness of reasoning chains, and reduces redundancy across modalities.

- Using our framework, we synthesize **NeSyGeo-Caption** and **NeSyGeo-CoT** training datasets with 100k samples, featuring high-quality images and reasoning chains, alongside a comprehensive geometric task evaluation set **NeSyGeo-Test** with 2,668 samples. These datasets are characterized by their diversity, rigor, and balanced distribution of information across image and text modalities.

- We demonstrate significant performance improvements on several MLLMs across multiple benchmarks using both RL and SFT training methods with our training sets, validating the effectiveness of our framework and the high quality of our datasets.

## 2 RELATED WORKS

### 2.1 GEOMETRIC PROBLEM-SOLVING

Early approaches to geometric reasoning predominantly relied on symbolic solvers that used formal languages to tackle the tasks. For instance, Inter-GPS (Lu et al., 2021) and PGDP (Zhang et al., 2022) employed symbolic methods by manually crafting reasoning rules and symbolic representations for geometric entities. These systems typically transform visual input into symbolic forms through instance segmentation and apply theorem search to derive solutions. However, these methods lacked scalability due to their dependence on manually designed rules. Their inability to generalize beyond specific problem types further limited their universality across diverse geometric challenges.

The advent of MLLMs has shifted the paradigm toward data-driven geometric reasoning, leveraging their robust reasoning capabilities. Recent advancements include GeoDRL (Peng et al., 2023) and GeoGen (Pan et al., 2025). Despite these developments, geometric reasoning poses significant challenges for MLLMs, requiring image perception, geometric knowledge, and multi-step reasoning. GeoSense (Xu et al., 2025) identifies the identification and application of geometric principles as a persistent bottleneck. Similarly, GeoEval (Zhang et al., 2024b) reveals that current MLLMs exhibit significantly low accuracy when facing more challenging geometric problems. MathVerse (Zhang et al., 2024c) further highlights MLLMs' over-reliance on textual information, underscoring the critical need for balanced multimodal datasets to enhance cross-modal reasoning capabilities.

### 2.2 MULTIMODAL GEOMETRY DATASETS

Large-scale, high-quality datasets are essential for enhancing the performance of MLLMs in solving geometric problems. Early datasets such as GeoS (Seo et al., 2015) (186 problems), Geometry-3k (Lu

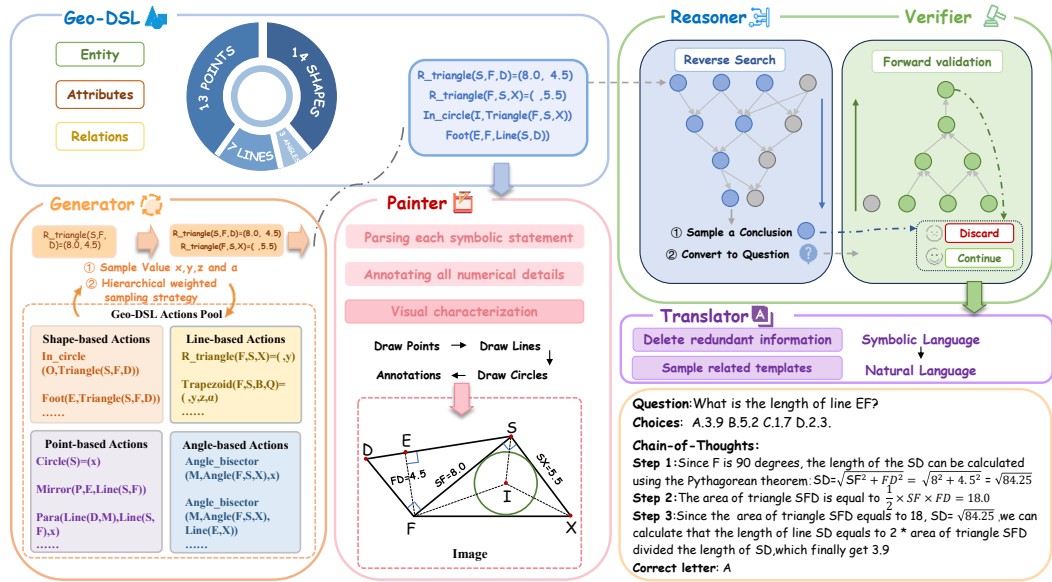

Figure 3: Our pipeline is centered around a symbolic language Geo-DSL. The Generator synthesizes a sequence in this language, from which Reasoner and Verifier produce logically sound Q&A pairs and reasoning chains. Subsequently, Painter and Translator render the symbolic core into semantically orthogonal visual and textual outputs.

et al., 2021) (3000 problems) and GeoQA (Chen et al., 2021) (4998 problems) utilized human manual annotation. Their datasets are thus limited to a small scale. With the development of MLLMs, datasets of greater magnitude have become essential. To address this, numerous efforts have shifted toward automatic data generation.

G-LLaVA (Gao et al., 2025) rephrased questions from GeoQA and Geometry-3k to create 115,000 Q&A pairs, but failed to enhance image variety. Template-based methods (Zhang et al., 2024d; Deng et al., 2024) typically rely on 10–20 predefined geometric figures, limiting the diversity of the generated images. AlphaGeometry (Trinh et al., 2024), a notable work that combines symbolic solvers for geometric proofs, employs a symbolic language definition. Yet, due to the absence of numerical attributes such as angle measures and segment lengths in its geometric space, attempts to automatically generate datasets using the AlphaGeometry framework.On the other hand, (Huang et al., 2025) and (Zhang et al., 2024a) are confined to caption datasets, failing to produce the numerical Q&A pairs critical for current MLLMs training. In contrast to prior approaches, our method pioneers a neuro-symbolic framework, being the first to integrate the precision of symbolic definition with the diversity of neural search for generating multimodal reasoning data.

## 3 METHODS

The overall framework of NeSyGeo is illustrated in Figure 3 .The pipeline is built upon **Geo-DSL**(Section 3.1), with generation process unfolds in three parts:

First, a symbolic **generator** performs action augmentations within the finite symbolic space to produce a Geo-DSL sequence. This approach effectively constrains the synthesis and augmentation process, ensuring validity and controllability by generating outside the infinite domains of natural language and image spaces (Section 3.2).

Second, to get questions with reasoning paths, we utilize **Reasoner** and **Verifier** with a two-stage generation process. The reverse search primarily ensures the diversity of the Q&A pairs, while the forward verification confirms the correctness of the CoT and the final answer. (Section 3.3)

Third, **painter** and **translator** map the generated Geo-DSL sequences back into natural language descriptions and visual image respectively. This process synthesizes high-quality images and valid text while avoiding intermodal information overlap. (Section 3.4)

## 3.1 Symbolic Definition

Existing symbolic languages present significant limitations. The syntax of AlphaGeometry (Trinh et al., 2024), for example, is tailored for formal proofs and omits numerical specifications. In contrast, InterGPS (Lu et al., 2021) suffers from excessive fragmentation, requiring multiple statements to define single elements, which substantially complicates the parsing and conversion process.

To surmount these challenges, we introduce **Geo-DSL**, a concise and comprehensive symbolic language for plane geometry. It is architected around an entity–attributes-relations framework, which we denote as a triplet $\mathcal{G} = (\mathcal{E}, \mathcal{A}, \mathcal{R})$, representing entities, attributes, and relations respectively. The syntax supports three primary declarative constructs: (i) *Attribute Assertions* $(e \rightarrow A_e)$, which bind quantitative properties to an entity; (ii) *Relational Assertions* $((e_1, \ldots, e_n) \rightarrow r)$, which define relationships among entities; and (iii) *Hybrid Declarations* that compose the two.

**Geo-DSL** offers two key advantages. First, it achieves descriptive completeness through a structured ontology that classifies all geometric primitives into four fundamental types (points, lines, angles, and shapes), representing the plane geometry with only 37 core statements. Second, its syntactic parsimony, where each entity is unambiguously instantiated via a single statement, enables both the principled composition of the symbolic action space and its deterministic parsing by our conversion engine. This synthesis of expressive power and structural minimalism renders **Geo-DSL** a robust and scalable framework for automated geometric representation and processing.

## 3.2 Symbolic Sequence Generation

**Generator:** To generate a Geo-DSL sequence, we employ a step-action augmentation procedure that iteratively synthesizes a sequence of statements, as detailed in Algorithm 1. Our methodology is initialized by configuring hyperparameters based on dataset characteristics: the sequence length $N$, weight matrices $\mathbf{I}$ and $\mathbf{A}$ that parameterize the selection distributions for geometric elements and symbolic actions (see Appendix H.2 for action details), and the value ranges $[l_{\min}, l_{\max}]$ and $[\theta_{\min}, \theta_{\max}]$ for lengths and angles, respectively.

The augmenter then commences an iterative synthesis over $N$ steps. At each step, after sampling continuous parameters $(x, y, z, \alpha)$ and assigning randomized labels to the geometric vertices, our generative process selects a discrete element-action pair $(v_j, a_k)$ using a hierarchical weighted sampling strategy.

First, an element $v_j$ is sampled from the available set $f_v$. The probability is proportional to its weight from the matrix $\mathbf{I}$, modulated by an element selection temperature $\tau_e$:

$$P(v_j) = \frac{(\mathbf{w}_I(v_j))^{1/\tau_e}}{\sum_{j' \in f_v} (\mathbf{w}_I(v_{j'}))^{1/\tau_e}} \tag{1}$$

Subsequently, conditioned on the chosen element $v_j$, a compatible action $a_k$ is sampled from the valid set $\mathcal{A}(v_j)$, with its probability determined by weights from the matrix $\mathbf{A}$ and modulated by an action selection temperature $\tau_a$:

$$P(a_k|v_j) = \frac{(\mathbf{w}_A(a_k, v_j))^{1/\tau_a}}{\sum_{k' \in \mathcal{A}(v_j)} (\mathbf{w}_A(a_{k'}, v_j))^{1/\tau_a}} \tag{2}$$

The newly formed statement $s_{\text{new}}$ is then integrated into the sequence $f_s$. This two-stage, temperature-controlled probabilistic approach guarantees the validity of each statement while promoting significant diversity in the generated sequences.

## 3.3 CoT Generation

To generate Q&A pairs accompanied by reasoning paths, we first formalize the underlying deductive task. The objective is to navigate from the initial premises to a valid conclusion through a sequence of logical steps. This process can be conceptualized through the formulation of a reasoning graph:

$$\mathcal{G} = (\mathcal{S}, \mathcal{R}, \hookrightarrow)$$

where:

- $\mathcal{S}$ is the set of all possible states, where each state $s \in \mathcal{S}$ represents the initial state or a deduced conclusion within the reasoning process.

- $\mathcal{R}$ denotes the set of geometric rules, with each rule $r \in \mathcal{R}$ encoding a principle of logical deduction that governs the relationships between states.

- $\hookrightarrow \subseteq \mathcal{S} \times \mathcal{R} \times \mathcal{S}$ is the state transition relation. A transition $(S_r, r, s')$ signifies that state $s'$ is derived from a subset of states $S_r \subset \mathcal{S}$ by applying rule $r$.

However, a direct implementation of this formalism via a deterministic inference engine is fraught with significant limitations. Firstly, such an engine is **computationally prohibitive**, as it necessitates an exhaustive enumeration of the combinatorial space of conditions at each intermediate state. Given a state characterized by a set of $N$ known conditions, the engine must evaluate the applicability of $C$ fundamental axioms and theorems. For each rule $i$, which is predicated on matching a specific set of $c_i$ conditions, the search space grows combinatorially, with a complexity lower-bounded by $\mathcal{O}\left(\sum_{i=1}^{C} \binom{N}{c_i}\right)$, rendering any exhaustive search intractable for non-trivial geometric configurations. Secondly, this approach exhibits **profound domain specificity**. The reasoning framework and its constituent logical rules are intrinsically hard-coded for plane geometry. This tight coupling to a single domain severely curtails the model's generalizability and fundamentally impedes its application for synthesizing datasets in other domains without substantial, non-trivial re-engineering.

To overcome these limitations, we pivot to leveraging the capabilities of LLMs, which can harness their extensive pre-trained knowledge and implicit intuitive faculties to facilitate rapid node expansion within the reasoning graph, obviating the need for bespoke rule engineering and thus offering a more agile and convenient paradigm. However, modern LLMs still face challenges when tackling intricate geometric tasks, and employing them for a direct, one-shot generation of Q&A pairs can result in erroneous information. To mitigate these issues, we introduce a novel two-stage methodology, which comprises a progressive reverse-search reasoner and a forward-validation verifier.

**Reasoner.** We introduce a novel search paradigm that operationalizes deductive reasoning as a Breadth-First Search (BFS) over the deductive lattice, accelerated by the learned preferences of LLMs. Instead of a uniform exploration, we leverage the reasoner to steer the expansion of the search frontier. From any given state $S_t$, the model $M_\theta$ is prompted to infer a single, contingent, and computable geometric conclusion $c_t$. This new conclusion augments the current state, effecting a state transition defined by $S_{t+1} = S_t \cup \{c_t\}$, where $c_t \leftarrow M_\theta(S_t)$. This iterative expansion establishes a form of progressive supervision and effectively prunes the vast search space, ensuring the fidelity and diversity of the resultant reasoning trajectories. By decomposing the complex, long-horizon inference task into a sequence of incremental, verifiable deductions, it substantially lowers the reasoning complexity and mitigates the model's propensity for hallucination. The terminal conclusions derived from this process are then randomly sampled to form candidate Q&A pairs.

**Verifier.** To verify the logical integrity of the generated Q&A pairs and to articulate a step-by-step CoT reasoning process. The questions generated by Reasoner, along with the corresponding conditions, are supplied as input. Verifier is prompted to produce both a final answer and its reasoning trajectory. These outputs are then cross-referenced against the answers derived from Reasoner. Only those pairs exhibiting concordance are retained for inclusion in our final dataset. The dual-verification protocol not only guarantees the correctness of the answers but also yields diverse and valid CoT narratives without necessitating an exhaustive traversal of the entire reasoning space.

### 3.4 INFORMALIZATION

The formal symbolic sequences from our Geo-DSL must be mapped back into the visual and textual modalities. To compel cross-modal reasoning, we enforce a principle of **information orthogonality**: All explicit constraints and premises $I$ in initial state $S_0$ are partitioned into disjoint visual modality $I_v$ and textual modality $I_t$, such that $I_v \cap I_t = \emptyset$.

**Painter.** For the visual modality, painter parses each Geo-DSL statement as a geometric primitive, deterministically rendering high-fidelity images via Matplotlib, which operates through two key phases: **Deterministic Coordinate Instantiation**, where the engine solver anchors a primary entity to the origin and sequentially propagates algebraic constraints to compute the exact spatial coordinates of all dependent elements; and **Robust Visual Instantiation**, which performs viewport normalization to shift the geometry into the positive quadrant while applying stochastic strategies for global scales and translations to mitigate spatial overfitting. Throughout this process, the rigorous symbolic space ensures the determinacy and uniqueness of the conversion. Crucially, to mitigate

inter-modal information imbalance, Painter employs a visual-centric annotation strategy, which involves embedding detailed information $I_v$ within the image space that are explicitly withheld from the accompanying textual description. Please refer to Appendix A.2 for more specific details. This design choice robustly compels MLLMs to intrinsically leverage visual cues during problem-solving, thereby fostering enhanced visual perception and facilitating the extraction of perceptually-grounded information.

**Translator.** Translator converts each symbolic statement to multiple predefined natural language templates. To ensure diversity and randomness, each template is stochastically generated by GPT-4o and subsequently undergoes a manual curation process. This guarantees that every geometric element is associated with a minimum of eight corresponding descriptive templates. The final text condition avoids intermodal information redundancy, while diverse templates ensure the validity and diversity of the synthesized condition text.

# 4 EXPERIMENTS

## 4.1 EXPERIMENTAL SETUP

**Dataset.** We generate a NeSyGeo-CoT dataset with 30k Q&A pairs and a NeSyGeo-Caption dataset with 70k Q&A pairs. The settings of generation process can be found in Appendix C.1. Additional dataset statistics and features are provided in Appendix B.1.

**Evaluation.** Our evaluation is conducted on several benchmarks: the Test set of GeoQA(Chen et al., 2021), the Test MINI set of MathVision(Wang et al., 2024b), and the MathVerse(Zhang et al., 2024c). For the MathVerse benchmark, we select the Vision Only, Vision Dominant, and Vision Intensive sets to better assess the visual perception and logical reasoning capabilities. We extract in-domain metrics from other datasets, including angle, area, length, and Plane Geometry, to effectively evaluate the models' capabilities in geometric reasoning problems, in addition to the GeoQA dataset, which focuses entirely on plane geometry. For GeoQA, we employed hard-coded extraction for comparison, while other evaluations are assessed using the automated VLMEvalKit framework(Duan et al., 2024). Appendix C provides additional experimental details and evaluation results.

## 4.2 EMPIRICAL RESULTS

**Quality: Manually evaluated by human experts, our dataset was found to contain high-quality CoT rationales, accurate Q&A pairs, and aesthetically pleasing images.**

To assess our dataset, we conducted a human expert evaluation with 40 experts (all at university undergraduate level or higher), comparing our NeSyGeo-CoT against Geo170K(question-answer pairs) proposed by G-LLaVA (Gao et al., 2025) and MAVIS-Instruct proposed by MAVIS(Zhang et al., 2024d). The evaluation was structured around five dimensions comprising nine metrics: Image, Question (including conditions), Reasoning path, Overall Impression and Final Answer. Each metric was rated on a 5-point scale. More details about our settings can be found in appendix C.2. NeSyGeo achieves state-of-the-art results on most metrics, ensuring a high degree of correctness, quality, and aesthetics, as corroborated by human evaluation.

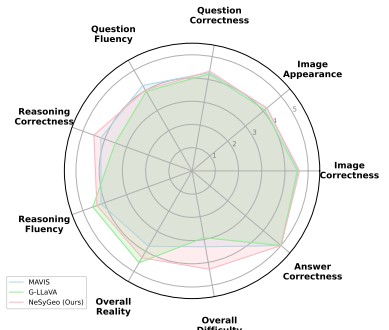

Figure 4: Human evaluation results comparison.

To validate our two-stage generation strategy on CoT and Q&A quality, we also performed an ablation study on the Reasoner and Verifier. As shown in Table 1, we generated 50 instances for each configuration. The results demonstrate that the Reasoner successfully expands the deductive search space and reduces reasoning hallucinations, which ultimately enhances both the problem difficulty and the depth of the reasoning chains. Concurrently, the Verifier provides effective supervision that is instrumental in ensuring the logical soundness and high accuracy of the final data.

**Performance: Our dataset enhances the geometric reasoning capabilities of models through both RL or SFT. Under the same data budget, ours is better than that from popular automatic generation frameworks.**

Table 1: Ablation study on the effects of the Reasoner and Verifier. Pass Rate measures the proportion of solutions accepted by the Verifier, while Pass@1 Rate (GPT-4o-mini's) serves as a proxy for problem difficulty. All other metrics align with our human preference experiments.

| Configurations | | Pass Rate | CoT-Quality | CoT-Accuracy | Ans-Correctness | Pass@1 Rate |
|---|---|---|---|---|---|---|
| Reasoner | Verifier | | | | | |
| **X** | ✓ | 78.0 | 4.34 | 4.66 | 5.00 | 68.0 |
| ✓ | **X** | — | 4.40 | 4.22 | 4.42 | 52.0 |
| ✓ | ✓ | 64.0 | 4.68 | 4.54 | 5.00 | 48.0 |

Table 2: **RL performance comparison**: Models trained with only 4k samples of **NeSyGeo-CoT** show performance gains over the base models, with the InternVL2.5-4B model exceeding the 8B variant in geometry problem-solving.

| Model | GeoQA | MathVision | | | MathVerse | | | |
|---|---|---|---|---|---|---|---|---|
| | | Angle | Area | Length | Angle | Area | Length | Plane |
| Qwen2.5-VL-3B | 53.3 | 26.3 | 26.3 | 21.1 | 31.3 | 20.9 | 37.0 | 32.5 |
| **Qwen2.5-VL-3B+Ours** | 55.7 (+2.4) | 26.3 (+0.0) | 42.1 (+15.8) | 26.3 (+5.2) | 32.6 (+1.3) | 23.5 (+2.6) | 37.2 (+0.2) | 35.5 (+3.0) |
| InternVL2.5-4B | 61.9 | 36.8 | 31.6 | 26.3 | 31.5 | 22.7 | 31.9 | 30.7 |
| InternVL2.5-4B+MAVIS | 63.5 (+1.6) | 31.6 (-5.2) | 26.3 (-5.3) | 31.6 (+5.3) | 37.1 (+5.6) | 20.9 (-1.8) | 35.3 (+3.4) | 33.7 (+3.0) |
| InternVL2.5-4B+R-CoT | 63.3 (+1.4) | 31.6 (-5.2) | 31.6 (+0.0) | 21.1 (-5.2) | 31.2 (-0.3) | 18.3 (-4.4) | 34.3 (+2.4) | 28.7 (-2.0) |
| **InternVL2.5-4B+Ours** | 69.2 (+7.3) | 42.1 (+5.3) | 36.8 (+5.2) | 26.3 (+0.0) | 39.9 (+8.4) | 24.9 (+2.2) | 36.1 (+4.2) | 36.7 (+6.0) |
| InternVL2.5-8B | 66.2 | 36.8 | 36.8 | 21.1 | 36.9 | 23.1 | 34.8 | 36.6 |

We first sample 4k samples from our NeSyGeo-CoT dataset and apply the Group Relative Policy Optimization (GRPO) algorithm to train two epochs with Deepseek R1's format and answer rewards. As shown in Tables 2, InternVL2.5-4B significantly improved in the angle knowledge domain with gains of 8.4 (MathVerse), 7.3 (GeoQA), and 5.3 (MathVision). Qwen2.5-VL-3B achieved +15.8 performance boost in the area domain of MathVision. Notably, across all evaluated metrics, the InternVL2.5-4B model trained on the NeSyGeo dataset achieves performance on par with or superior to its 8B counterpart.

To ensure a fair comparison with others, we randomly sampled same budget from MAVIS-Instruct (Zhang et al., 2024d) and TR-GeoMM proposed by TR-CoT (Deng et al., 2025) and maintained consistent settings. As illustrated in Figure 5 and Table 2, while all datasets help the model improve over the baseline, our dataset outperformed others in most metrics, validating its superior performance.

Table 3: **SFT performance comparison**: Compared with baseline, Qwen2.5-VL-7B model trained in our datasets demonstrates performance improvements on most metrics within GeoQA and MathVerse benchmarks

| Model | GeoQA | MathVerse | | | |
|---|---|---|---|---|---|
| | | Angle | Area | Length | Plane |
| Qwen2.5-VL-7B | 69.4 | 43.0 | 27.5 | 46.2 | 44.1 |
| **Qwen2.5-VL-7B+Ours** | 71.8 (+2.4) | 46.1 (+3.1) | 23.1 (-4.4) | 49.5 (+3.3) | 46.7 (+2.6) |
| LLaVA-7B | 22.6 | 28.5 | 6.6 | 16.5 | 20.4 |
| **LLaVA-7B+Ours** | 26.1 (+3.5) | 30.6 (+2.1) | 7.7 (+1.1) | 19.2 (+2.7) | 22.9 (+2.5) |

Table 4: **Scalability Analysis**: Performance on GeoQA with varying dataset ratios.

| Training Data | GeoQA |
|---|---|
| Baseline | 69.4 |
| 0% | 72.3 |
| 30% | 72.8 |
| 50% | 74.3 |
| 80% | 76.0 |
| 100% | 77.2 |

We also conducted SFT experiments, initially training on our NeSyGeo-Caption dataset to enhance the models' perception of geometric images, followed by training on the NeSyGeo-CoT dataset to improve reasoning capabilities. Evaluation results on MathVerse (Vision Intensive) and GeoQA are presented in Table 3. The trained model demonstrates performance improvements over the base model on most metrics.

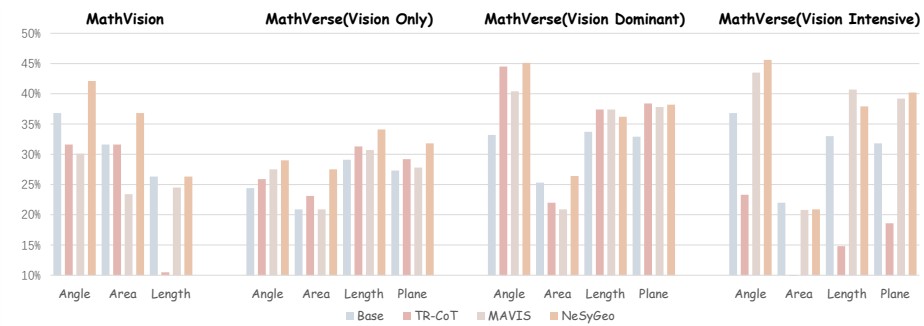

Figure 5: Efficiency comparison of our NeSyGeo-CoT dataset versus other mainstream automated synthesis methods. The models are trained using RL methods with InternVL2.5-4B.

To investigate the scalability of our framework, we performed supervised fine-tuning on the Qwen2.5-VL-7B-Instruct model using varying scales of our generated NeSyGeo-CoT dataset. Specifically, we constructed five distinct training sets by merging the Geo170K instruction-tuning split (117k samples) with random subsets of our data corresponding to 0%, 30%, 50%, 80%, and 100% of the total volume. The results, as illustrated in Table 4, highlight the promising scalability of our method. We observe a consistent improvement in model accuracy as the quantity of synthetic data increases, thereby demonstrating the robustness and generalizability of our approach.

**Generability: The performance gains of models are from true reasoning, not just distilling solutions leaked from contaminated data**

Although we have implemented stringent safeguards to ensure that our datasets do not access the test set, a critical inquiry is whether the efficacy of our framework stems from inadvertent data leakage introduced during the generation process. Adopting the methodology delineated by Wu et al. (Wu et al., 2025), we conducted an experiment utilizing the GeoQA test set.

For each question, we furnished the models with partial prompts, specifically truncated to the initial 40%, 60%, and 80% of their original length. We then evaluated two key metrics: (1) The ROUGE-L score, quantifying the textual overlap between the model's generated output and gpt-4o's output, and (2) The final answer accuracy. This empirical evaluation was performed on two distinct models, Qwen2.5-VL-3B and InternVL2.5-4B.

While a marginal increase in certain ROUGE-L scores as shown in Table 5, we posit that this is a manifestation of the model's augmented reasoning capabilities, which empower it to perform more structured and relevant explorations from a limited context. The consistently negligible accuracy scores across all truncated inputs strongly indicate that the models have not merely memorized the test instance. Consequently, we conclude that there is no substantive evidence to suggest that the model's performance gains are attributable to data contamination.

Table 5: Analysis of Potential Data Contamination using Truncated Prompts.

| Model | 0.4 Ratio | | 0.6 Ratio | | 0.8 Ratio | |
|---|---|---|---|---|---|---|
| | ROUGE-L | Acc. | ROUGE-L | Acc. | ROUGE-L | Acc. |
| QwenVL-base | 28.04 | 0.00 | 29.88 | 0.00 | 31.95 | 0.00 |
| QwenVL-RL | 21.29 | 0.00 | 36.72 | 0.00 | 40.57 | 0.00 |
| InternVL-Base | 26.82 | 0.00 | 40.04 | 0.00 | 48.07 | 4.69 |
| InternVL-RL | 30.00 | 0.00 | 44.91 | 0.00 | 57.37 | 1.54 |

**Visual Perception: Models trained on NeSyGeo data shows modest gains over information-redundant datasets, yet achieves substantial improvements when image understanding is necessary. This indicates that ours enhance model's cross-modal reasoning.**

A key question is whether reducing redundancy and forcing models to extract visual information improves their geometric reasoning capabilities. We evaluate models on the MathVerse(Text Domi-

nant), which provides redundant text descriptions and implicit properties enabling reasoning without images, and the MathVerse(Vision Only) version, where all information is embedded entirely within the images. For this comparison, we selected two text-redundant datasets: NeSyGeo+RED, the original NeSyGeo-CoT dataset supplemented with textual equivalents of its image annotations, and the R-CoT dataset. Results presented in Table 5 show that our model outperforms the baseline on Text-Dominant but lags behind other datasets in some metrics. On Vision-Only, our model surpasses them across all metrics, demonstrating enhanced geometric reasoning and visual perception.

## 5 CONCLUSION

This paper introduces NeSyGeo, a neuro-symbolic framework for automatically synthesizing multimodal geometric datasets. Our approach transforms the generation process into a controllable symbolic space using Geo-DSL, utilizes Reasoner and Verifier for reverse search and forward validation to produce Q&A pairs, and then maps the symbolic representation back to image and natural language spaces via Painter and Translator. Using this framework, we construct NeSyGeo-CoT and NeSyGeo-Caption datasets, totalling 100k samples. We also propose NeSyGeo-Test, a comprehensive benchmark for evaluating MLLMs' geometric reasoning capabilities. Our datasets significantly and consistently improve the reasoning abilities of multiple MLLMs through both SFT and RL.

## ETHICS STATEMENT

NeSyGeo datasets presented in this paper are generated entirely through a synthetic, neuro-symbolic framework. The core generation process does not involve any user data, personally identifiable information, or web-scraping. Human involvement was limited exclusively to a voluntary study for evaluating the final dataset's quality. All data collected from these human experts was fully anonymized to ensure strict confidentiality and the detailed evaluation protocol is provided in Appendix C.2. This methodology mitigates concerns related to both data privacy and social bias. Given the highly specialized nature of this mathematical task, we assess the potential for misuse or negative societal impact to be minimal. We release our contributions to the research community to responsibly foster progress in automated scientific reasoning.

## REPRODUCIBILITY STATEMENT

To ensure the reproducibility of our research, we provide a comprehensive description of our data generation methodology throughout the main paper and in the Appendix A. Each module of our synthesis pipeline is delineated with its corresponding parameters and implementation details. We commit to making the complete synthetic dataset publicly available to the research community upon publication to facilitate further investigation and development. Furthermore, all hyperparameters and key configurations for our model training and experiments are fully disclosed in the Appendix C.

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

APPENDIX CONTENTS

# A MORE DETAILS OF METHODS

## A.1 GENERATOR

As outlined in Algorithm 1, for each step, we first generate three lengths, $x$, $y$, and $z$, along with an angle $\alpha$, sampled from predefined ranges. We then randomly selecting vertex labels. Subsequently, based on the weight matrices $A$ and $I$, we determine the specific statement to be selected. The chosen statement, paired with its corresponding numerical values, is then appended to the Geo-DSL sequence.

---

**Algorithm 1** The overall framework of the symbolic sequence generation process

---

    **Input:** Step count $N$, Action weight matrix $A$, Element selection weight matrix $I$, Line length range $[l_{min}, l_{max}]$, Angle range $[\theta_{min}, \theta_{max}]$.          ▷ Set customizable hyperparameter
    **Output:** Generated Geo-DSL statement sequence $f_s$.
1: Initialize $f_s$=Initialize( ) .          ▷ Initialize the sequence $f_s$ with the first statement
2: Initialize symbolic state space elements $f_v = $ Initialize($f_s$).      ▷ Initialize state based on $f_s$
3: **for** $i = 1$ to $N$ **do**
4:      Randomly sample $x, y, z$ from $[l_{min}, l_{max}]$.
5:      Randomly sample $\alpha$ from $[\theta_{min}, \theta_{max}]$.
6:      Element $v_j$=Selected_elements($f_v$,$I$).          ▷ Select element $v_j$ from $f_v$ randomly.
7:      Action $a_k$=Selected_action($v_j$,$A$).      ▷ Select action $a_k$ based on the type of $v_j$ randomly
8:      $s_{new}$=Generate_DSL($a_k$, x, y, z, $\alpha$).          ▷ Generate new DSL statement
9:      $f_s$=Update($f_s$,$s_{new}$)          ▷ Add the new statement to the sequence
10:      $f_v$=Update($f_v$,$s_{new}$).          ▷ Update state space elements
11: **end for**
12: **return** $f_s$.

---

## A.2 PAINTER

The Painter generates geometric diagrams from using a two-stage pipeline, which first computes a metrically-precise coordinate representation and then renders it using a robust, annotation-aware engine.

## A.3 PAINTER: THE COMPUTATION-INSTANTIATION PIPELINE

To manage the translation from the symbolic Geo-DSL to high-fidelity visual representations, we implement a rigorous Computation-Instantiation Pipeline.

### A.3.1 DETERMINISTIC COORDINATE INSTANTIATION

The painter initiates by "anchoring" the primary geometric entity (e.g., Point $A$) to the coordinate origin $(0, 0)$. Subsequent DSL statements (e.g., Triangle(A,B,C)=$(x,y,\alpha)$) are treated as a system of algebraic and geometric constraints. Leveraging the anchored reference, the solver propagates these constraints like trigonometric functions and affine transformations to compute the exact coordinates of dependent elements (e.g., $B$ and $LineCD$).

This process iterates sequentially until the spatial position of every geometric element is resolved. The result is a unique, geometrically precise numerical materialization of the symbolic DSL. The stage culminates in a Geometric Primitive Dictionary that encodes the precise coordinates, attributes, and topology of all geometric entities, which serves as the requisite input for the subsequent rendering engine and visual annotation protocols.

### A.3.2 ROBUST VISUAL INSTANTIATION

The constraint solving in the previous phase may yield coordinates outside the canonical first quadrant. To mitigate rendering artifacts, the engine performs a pre-computation of the precise position for all entities. It then applies a global translation vector to the entire geometric constellation, shifting it

into the positive coordinate quadrant as a viewport Normalization. This ensures absolute robustness against clipping or coordinate misalignment.

To further enhance visual diversity and aesthetic fidelity, Painter incorporates stochastic rendering strategies. During each synthesis instance, the global scale (pixel-to-unit ratio) is stochastically sampled, and the geometric constellation is randomly translated within the viewport. Furthermore, distinct categories of geometric primitives are rendered using a diverse color palette to optimize visual discriminability. These randomization procedures guarantee robust data variance, effectively mitigating the risk of the model overfitting to superficial stylistic biases such as absolute position or scale.

### A.3.3 VISUAL-CENTRIC ANNOTATION STRATEGY

To maximize the interpretability of the generated diagrams, Painter enforces a strict set of visual encoding rules designed to embed explicit semantic information directly into the pixel space:

- **Segment Length:** Every line segment is explicitly labeled with its identifier and corresponding numerical length, rounded to one decimal place for clarity.

- **Angle Annotations:** The measure of any non-right angle is inscribed directly within the angle's arc. As noted in the rendering phase, right angles are exclusively denoted by the conventional square symbol.

- **Circle Radius:** Circle radii are represented by a dashed line extending from the center to the circumference, explicitly marked with the label "$r = [length]$".

Painter not only ensures the diversity and rigorous precision of the dataset but also enforces a unique cross-modal interaction regime. Crucially, to prevent information redundancy and compel a true synthesis of visual and textual data, any property that is graphically rendered by Painter is intentionally omitted from the accompanying text description.

### A.4 TRANSLATOR

To provide the Reasoner with sufficient context for its reasoning process, our Translator module initially generates a complete and comprehensive set of textual conditions. However, to mitigate information inconsistency across modalities in the final synthesized dataset, the Translator performs a crucial refinement step. It systematically prunes any explicit constraint-based information that is visually rendered in the image (e.g., numerical values for segment lengths or angles). This process prevents the model from developing "shortcuts" based on redundant textual data, which would impair its cross-modal reasoning capabilities. The final text exclusively retains abstract information about entities (e.g., "isosceles triangle") and their semantic relations (e.g., "point O is the incenter of triangle ABC"), compelling the model to ground its understanding in the visual domain.

### A.5 REASONER

We employ DeepSeek-R1 as our reasoning LLM to conduct a forward search process. At each step, the model receives the conditions of the current state and is prompted to infer a single, new contingent and computable geometric property or relationship that augments this state. The specific prompt used for this task is shown in Box A.5.

---

**The prompt for Reasoner to reverse search**

Use the mathematics you know to make simple and accurate inferences and get the conclusions based on image descriptions.
Focus on extracting as much information as possible. Keep the final answer of all reasoning to one decimal place. Use <conclusion> </conclusion> to include all your inferences.
Example is as follows:
Output: <conclusion> 1. The area of Triangle SFD is 18.0. 2. The length of line EF is 3.9. 3. The measure of angle DFS is 29.4°. </conclusion>

- - - - - - - - - - - - - - - - - - - - - - - - - - - - - - - - - - - - - - - - - - - - - - - - -

Here is the image description input: Current state

---

## A.6 VERIFIER

We employ DeepSeek-V3 to conduct the forward verification process. The prompt used for this task is detailed in Box A.6.

---

**The prompt for the Verifier to Forward Validation**

Here is a geometry problem with the following information:
Image description: Initial state
Question: Questions from Reasoner
Following the above condition and question, think step by step and answer the following question directly.
After thinking process, you should provide your final concise reasoning steps in <steps></steps> tags and your final answer in <answer> </answer> tags.
The answer to the question is only allowed to contain one number or angle.

---

## B   MORE DETAILS OF DATASETS

### B.1   STATISTICS OF DATASETS

Detailed numerical statistics and element distribution for the NeSyGeo-Caption and NeSyGeo-CoT datasets are presented in Table 7 and 8. For element distribution statistics, we randomly sampled 1.8k Geo-DSL sequences corresponding to images from each dataset, counting the frequency of different geometric elements. To facilitate interpretation, these elements are converted into corresponding natural language descriptions.

### B.2   DIFFICULTY DEFINITIONS

To support evaluation and training paradigms such as curriculum learning, we annotate each sample with a difficulty level. Given that geometric reasoning tasks primarily require models' image perception and logical reasoning capabilities, we scientifically define the difficulty level as

$$\alpha \times \text{perception difficulty} + (1 - \alpha) \times \text{reasoning difficulty}, \tag{3}$$

where perception difficulty is the number of Geo-DSL statements, and reasoning difficulty is the number of reasoning steps. Here, we set alpha as 0.3.

Each synthesized sample includes detailed meta-information stored as a symbolic form based on our Geo-DSL language. This symbolic form accurately describes the geometric setup and offers promising directions for future research. For instance, valid geometric configurations could be generated by augmenting or mutating existing symbolic forms within constrained parametric bounds.

### B.3   DETAILS OF NESYGEO-TEST BENCHMARK.

Our NeSyGeo-Test benchmark comprises 2668 Q&A pairs. Consistent with the training set, numerical annotations are embedded in the image space, with only essential conditions and questions provided

Table 7: Statistics of NeSyGeo-Caption

| Statistic | Number |
|---|---|
| *Total Counts* | |
| Total number of images | 70k |
| Total number of captions | 70k |
| *DSL Statement Percentage* | |
| One Statement | 5.4% |
| Two Statements | 25.8% |
| Three Statements | 34.7% |
| Four Statements | 23.4% |
| Five Statements | 10.7% |
| *Length of Captions* | |
| Maximum length (words) | 220 |
| Minimum length (words) | 34 |
| Average length (words) | 73.3 |
| Average length (characters) | 385.4 |
| *Image Dimensions* | |
| Average dimensions (pixels) | $723.1 \times 724.0$ |

Table 8: Statistics of NeSyGeo-CoT

| Statistic | Number |
|---|---|
| *Total Counts* | |
| Total number of images | 15.3k |
| Total number of Q&A pairs | 30.1k |
| *Question Statistics* | |
| Length-based type | 54.6% |
| Area-based type | 34.4% |
| Angle-based type | 11.1% |
| Average length (words) | 26.9 |
| Average length (characters) | 140.6 |
| *CoT Statistics* | |
| Below four steps | 35.4% |
| Four steps or above | 64.5% |
| Average length (words) | 91.8 |
| Average length (characters) | 365.9 |
| *Image Dimensions* | |
| Average dimensions (pixels) | $731.0 \times 727.4$ |

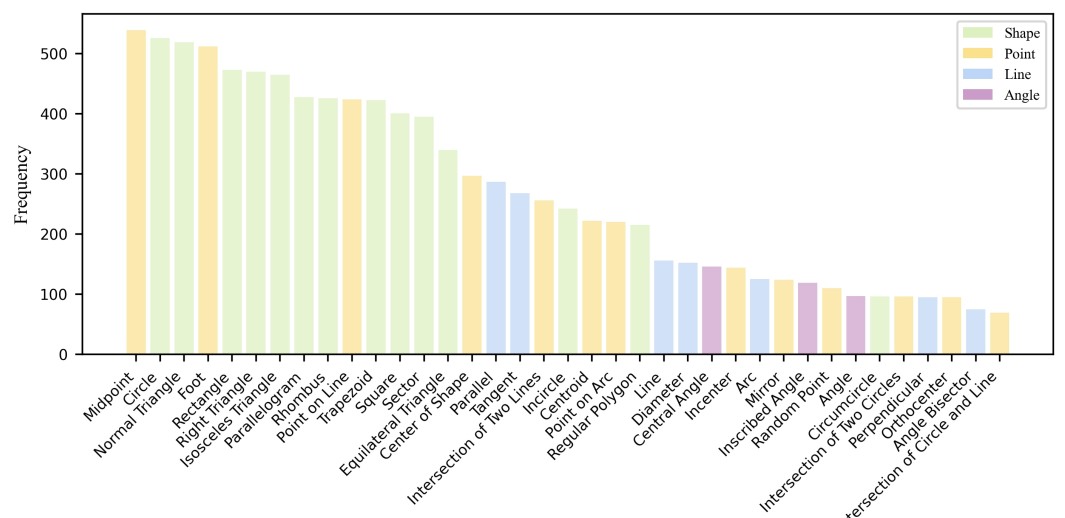

Figure 6: Frequency of different geometric elements. To facilitate interpretation, these elements are converted into corresponding natural language descriptions.

in the text. The type of numerical quantity categorizes the dataset sought: Angle (658 pairs), Shape (730 pairs), and Length (1280 pairs). Shape type includes shape area and perimeter, while length type includes edge and arc lengths. Based on the difficulty level in B.2, problem difficulty is divided into three levels: Easy (1537 pairs), Medium (908 pairs), and Hard (223 pairs).

# C MORE DETAILS AND RESULTS OF EXPERIMENTS

## C.1 SETTINGS OF DATASET GENERATION

To synthesize a diverse dataset, we set hyperparameters for generating images by configuring the step count, length range, and angle range. The step count $N$ ranges from one to four. The line length is defined within the basic range $[l_{min}, l_{max}] = [1, 5]$, scalable by any multiple of 2 (e.g., $[4, 20]$). The angle is constrained to multiples of $15°$ within $[15°, 165°]$, with increased weights assigned to special angles. The temperature hyperparameters $\tau_e$ and $\tau_a$ were set to a default value of 1. We generate various types of weight matrices $A$ and $I$ by adjusting their corresponding values.

## C.2 HUMAN PREFERENCE EVALUATION

To rigorously assess the quality of our generated data, we conducted a comprehensive human evaluation study. We compared our NeSyGeo-CoT dataset against two prominent, automatically generated geometry datasets: G-LLaVA and MAVIS.

**Methodology**

We recruited 40 volunteers. To ensure a proficient and consistent level of domain expertise, a prerequisite for participation was holding a university undergraduate degree or higher. To ensure the objectivity and accuracy of the evaluation, participants were informed that the study pertained to middle school mathematics education and that their primary task was to meticulously verify the correctness of the provided solutions and final answers for each problem. We guaranteed participants that their responses would be completely anonymized to protect their privacy, with the right to withdraw at any time without penalty.

Each participant was presented with a questionnaire containing nine geometric problems. To mitigate potential ordering effects and sampling bias, the problems were composed of three samples randomly and independently drawn from each of the three datasets being evaluated.

**Evaluation Criteria**

Participants were asked to rate each sample on a 5-point Likert scale across five key dimensions designed to capture the multifaceted quality of a multimodal reasoning problem:

**Image** This dimension evaluates the visual output. **Image Correctness** assesses whether the image 1)accurately reflects all entities and constraints from the text premise.2) has no conflict itself **Image Appearance** rates the human aesthetic quality, including the clarity of annotations and overall layout.

**Question** This assesses the generated question. **Question Correctness** evaluates its logical validity and solvability based on the provided context, while **Question Fluency** rates its grammatical structure and naturalness.

**Reasoning** This pertains to the generated solution. **Reasoning Correctness** scrutinizes the logical and mathematical validity of the step-by-step thought process. **Reasoning Fluency** assesses the clarity and readability of the explanation.

**Overall** This captures the evaluators' subjective impressions. **Overall Reality** gauges the plausibility of the problem, assessing if it feels like a genuine question from a textbook or exam. **Overall Difficulty** is the participant's rating of the cognitive challenge involved.

**Answer** **Answer Correctness** provides the definitive assessment of problem-solving capability by verifying if the final numerical answer derived from the reasoning process is objectively correct.

Results The aggregated results are presented in Picture 4. The findings indicate a clear preference for our NeSyGeo-CoT dataset. While all datasets scored perfectly on answer correctness—an expected outcome for synthetically generated data—our dataset was rated significantly higher in Reasoning Path Quality and Overall Difficulty. This suggests that our framework not only generates valid and high-fidelity problems but also produces content that is more complex and challenging, thereby validating its effectiveness for advancing the reasoning capabilities of MLLMs.

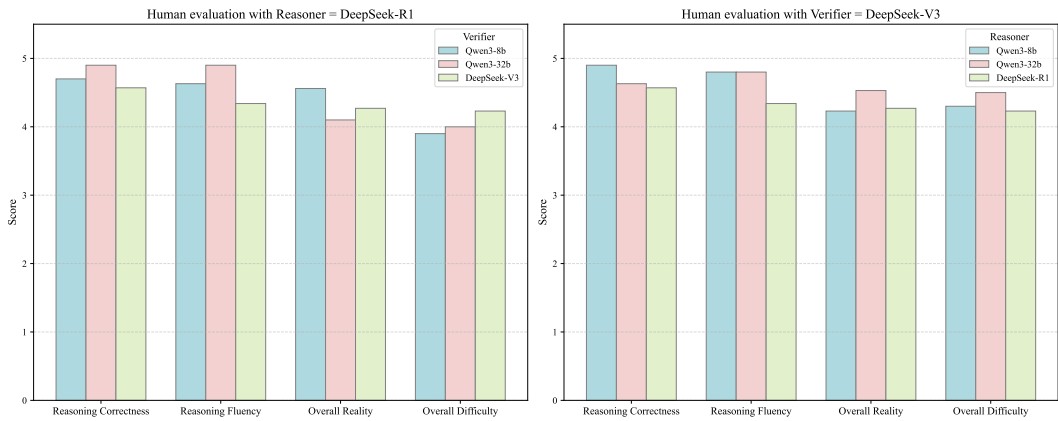

Figure 7: Ablation study of reasoner and verifier. The left panel illustrates performance metrics when Deepseek-R1 functions as the Reasoner, while the right panel presents the performance of Deepseek-V3 as the Verifier.

## C.3 ABLATION STUDY OF REASONER AND VERIFIER

To investigate the sensitivity of our framework to model capacity, we conducted an ablation study utilizing smaller open-source models, specifically Qwen3-8b and Qwen3-32b, functioning as the Reasoner and Verifier. For this analysis, 600 samples were generated for each configuration. These samples were subsequently evaluated by 10 domain experts following the evaluation protocol outlined in Appendix C.2

The empirical results indicate that employing smaller models (e.g., 8B) does not result in a sharp performance decline, which implies the inherent robustness of the proposed neuro-symbolic framework. However, comparative analysis reveals that the 8B model exhibits slightly lower scores in Overall Difficulty and Overall Reality metrics relative to larger models. Consequently, while the framework remains viable with smaller architectures, we recommend utilizing models at the 32B scale or larger to guarantee optimal data complexity and fidelity. Detailed visualizations of these comparative results are provided in Figure 7.

## C.4 EQUIVALENCE VALIDATION WITH SYMBOLIC SOLVERS

To demonstrate that the problems generated by our framework possess both provability and appropriate complexity, we conducted an external validation using established open-source symbolic solvers, specifically InterGPS (Lu et al., 2021) and Pi-GPS Zhao et al. (2025). It is important to note that AlphaGeometry was excluded from this comparative analysis because its underlying language and theorem rules do not support numerical metric calculations (e.g., area and length), which are essential for our dataset.

We sampled 20 questions from the NeSyGeo-Test dataset and manually translated the problem conditions and queries into the specific symbolic language formats required by the respective solvers. The quantitative results are presented in Table 9.

Table 9: Symbolic solver performance comparison on sampled problems.

| Method | Pass Rate | Average Steps |
|---|---|---|
| Inter-GPS | 20.00 | 31.50 |
| Pi-GPS | 30.00 | 5.33 |
| Human | 100.00 | 4.75 |

A detailed case analysis reveals that the failures are primarily attributable to the inherent limitations of the solvers themselves rather than the insolvability of the problems. Specifically, the failure breakdown is as follows: 20% due to missing theorems within the solver's knowledge base, 20% due

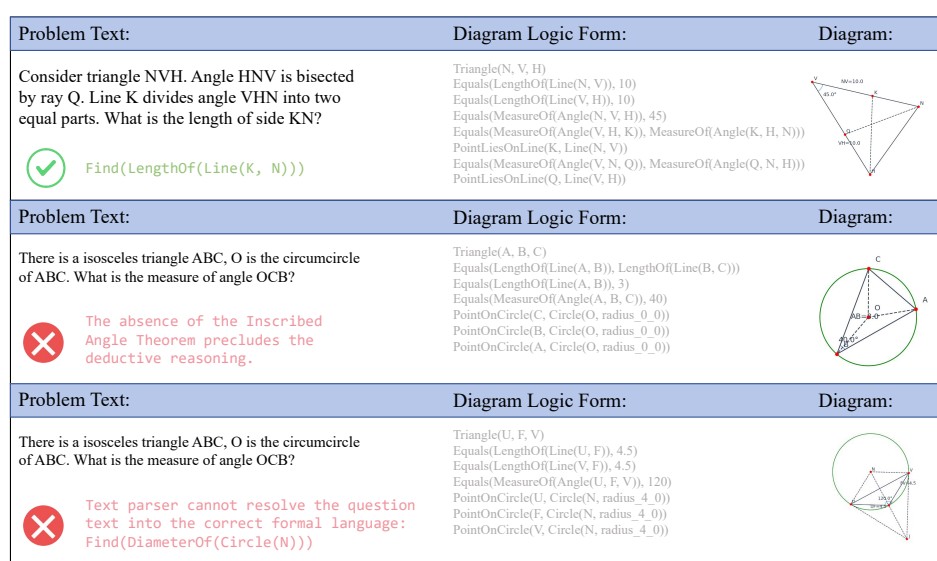

Figure 8: Selected samples from the NeSyGeo dataset solved by the InterGPS symbolic solver.

Table 10: Detailed RL experiments evaluation on MathVerse. Here, 'AGL', 'ARA', 'LTH', and 'PG' denote angle, area, length, and plane geometry, respectively.

| Model | Vision Intensive | | | | Vision Dominant | | | | Vision Only | | | |
|---|---|---|---|---|---|---|---|---|---|---|---|---|
| | AGL | ARA | LTH | PG | AGL | ARA | LTH | PG | AGL | ARA | LTH | PG |
| Qwen2.5-VL-3B | 31.6 | 22.0 | 34.6 | 33.3 | 31.6 | 17.6 | 40.7 | 31.4 | 30.6 | 23.1 | 35.7 | 32.7 |
| InternVL2.5-4B | 36.8 | 22.0 | 33.0 | 31.8 | 33.2 | 25.3 | 33.7 | 32.9 | 24.4 | 20.9 | 29.1 | 27.3 |
| InternVL2.5-8B | 44.0 | 23.1 | 36.3 | 41.8 | 40.4 | 20.9 | 36.8 | 37.3 | 26.4 | 25.3 | 31.3 | 30.6 |
| Qwen2.5-VL-3B+RL | 33.7 | 23.1 | 36.8 | 34.7 | 32.1 | 20.9 | 37.4 | 36.6 | 32.1 | 26.4 | 37.4 | 35.1 |
| InternVL2.5-4B+RL | 45.6 | 20.9 | 37.9 | 40.2 | 45.1 | 26.4 | 36.2 | 38.2 | 29.0 | 27.5 | 34.1 | 31.8 |

to failures in correctly parsing the search targets, and 40% due to internal solver errors. Furthermore, the relatively high number of steps required for symbolic solutions (particularly for Inter-GPS) further validates the complexity and high quality of our data. We provide detailed visualizations of these case studies in Figure 8.

## C.5    REINFORCEMENT LEARNING

We utilized the VLM-R1 (Shen et al., 2025) framework for RL experiments, conducted on 6 vGPU-32 GB. We set epochs to 2, num generations to 6, batchsize to 1. To enhance the visual perception capabilities of MLLMs, parameters of the language model and vision modules are set to be trainable.

Table 10 presents the detailed performance of models trained on various automatically synthesized datasets across the MathVerse benchmark. Models trained using our dataset demonstrate superior performance on most metrics compared to others, exhibiting substantial performance gains relative to the base model.As shown in Figure 9. We also evaluated model performance as RL training steps increased when using NeSyGeo-CoT. Most metrics improved with more training steps, demonstrating the robustness and effectiveness of our datasets.

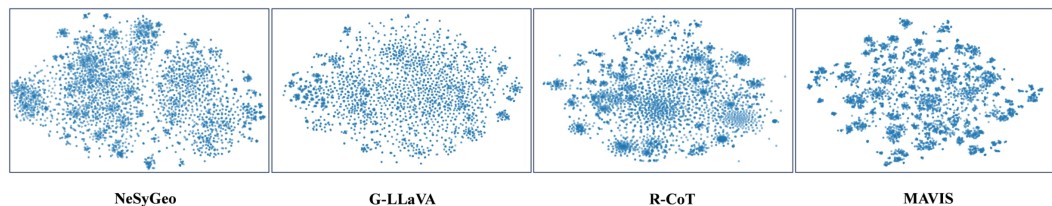

**NeSyGeo**   **G-LLaVA**   **R-CoT**   **MAVIS**

Figure 10: T-SNE of the text features of different automatic frameworks. The G-LLaVA method augments the text space on the manually annotated GeoQA dataset. Thus, its text diversity can approximate that of real data more closely. Similar to G-LLaVA, our method exhibits a uniform distribution in the space, demonstrating superior diversity.

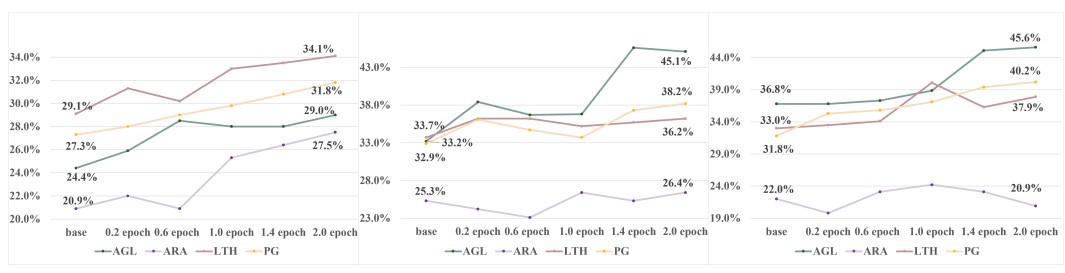

Figure 9: Model performance on Mathverse as the RL training steps increase. With InternVL2.5-4B as our base model, most metrics exhibit progressive improvement throughout training, demonstrating the robustness and effectiveness of our datasets.

### C.6 Supervised Finetuning

For SFT experiments, we employed the LLaMA-Factory (Zheng et al., 2024) framework on 2 A800 GPUs with LoRA. We set the learning rate of $1 \times 10^{-5}$, LoRA rank of 64, and use Adam optimization. Training on NeSyGeo-Caption used 1 epoch, while NeSyGeo-CoT used 2 epochs.

### C.7 Diversity of Dataset

**Diversity: Datasets synthesized by our NeSyGeo framework exhibit high diversity in text and visual features.**

A critical challenge for automatic data synthesis methods is whether the dataset is sufficiently diverse to avoid quality degradation due to potential overfitting risks from inherent domain constraints. We employ t-SNE (van der Maaten & Hinton, 2008) dimensionality reduction for mapping in text space to evaluate the diversity across different methods. This analysis allows us to assess the diversity of the textual descriptions themselves. Given that in geometric problems, the text conditions depict specific visual elements, and the diversity observed in the text space also serves as a valuable indicator of the diversity in the corresponding visual diagrams. To ensure a fair comparison, we remove all prompts related to guiding large models, retaining only condition and question texts, and randomly sample 5k texts from each dataset. The results are illustrated in Figure 10. Our method and G-LLaVA (Gao et al., 2025) exhibit uniformly distributed features in the space, indicating low data overlap and high diversity. In contrast, R-CoT and MAVIS display varying degrees of clustered distribution, indicating more feature-similar samples.

## D NeSyGeo-Test Benchmark

To probe the genuine cross-modal reasoning capabilities of current models, our empirical analysis is grounded in the NeSyGeo-test benchmark. This dataset was engineered to address the shortcomings

Table 11: NeSyGeo-Test Benchmark on several mainstream MLLMs. The highest accuracy for open-source and closed-source MLLMs is marked in red and blue, respectively.

| Model | Param | Task Type | | | Question Difficulty | | | Total |
| | | Angle | Shape | Length | Easy | Medium | Hard | |
|---|---|---|---|---|---|---|---|---|
| *Open-source MLLMs* | | | | | | | | |
| Qwen2.5-VL-3B-Instruct | 3B | 44.1 | 31.8 | 27.8 | 30.5 | 29.1 | 31.9 | 30.9 |
| Qwen2.5-VL-7B-Instruct | 7B | 38.3 | 30.7 | 32.2 | 36.8 | 27.5 | 32.3 | 33.3 |
| Qwen2.5-VL-32B-Instruct | 7B | 38.3 | 30.7 | 32.2 | 65.8 | 49.4 | 41.7 | 58.2 |
| InternVL2.5-8B | 8B | 55.9 | 56.2 | 55.4 | 64.5 | 49.3 | 43.3 | 55.8 |
| LLaVA-NeXT-7B | 13B | 15.7 | 15.6 | 16.0 | 15.8 | 16.0 | 15.2 | 15.8 |
| LLaVA-NeXT-13B | 7B | 23.7 | 15.5 | 15.4 | 18.6 | 14.7 | 13.3 | 16.4 |
| LLaVA-NeXT-34B | 34B | 23.7 | 21.0 | 18.5 | 19.1 | 21.5 | 19.2 | 19.9 |
| *Closed-source MLLMs* | | | | | | | | |
| InternVL3-latest | – | 81.7 | 65.2 | 68.3 | 77.8 | 62.0 | 56.7 | 68.7 |
| GPT-4o-mini | – | 58.7 | 62.1 | 55.7 | 63.0 | 54.0 | 41.7 | 58.2 |
| Claude-3.5-Sonnet-latest | – | 68.8 | 78.0 | 71.0 | 77.8 | 73.2 | 56.5 | 74.5 |
| Qwen-VL-plus | – | 38.5 | 29.6 | 31.7 | 36.6 | 27.2 | 29.6 | 32.8 |
| Gemini-2.0-Flash | – | 36.8 | 60.4 | 67.7 | 54.3 | 63.8 | 61.0 | 58.1 |

of its predecessors. It comprises high-fidelity, programmatically rendered images and mitigates modality imbalance by ensuring crucial information is distributed across both the visual and textual inputs. A cornerstone of its design is the elimination of textual redundancies, a feature that compels models to ground their deductions in the visual context and prevents reliance on unimodal heuristics. Evaluation results on current mainstream open-source and closed-source MLLMs are shown in Table 11.

## D.1 QUANTITATIVE PERFORMANCE

A key observation is the benchmark's considerable difficulty, which challenges even state-of-the-art MLLMs. This is particularly evident in the `Hard` difficulty tier, where the highest accuracy achieved by any closed-source model is merely 56.7%, while the leading open-source model, Qwen2.5-VL-32B-Instruct, scores only 43.3%. These results underscore the significant room for improvement in complex geometric reasoning. Furthermore, the effectiveness of our difficulty stratification is validated by the consistent performance degradation exhibited by most models as they progress from `Easy` to `Medium` and `Hard` questions. This trend confirms the rational design of our benchmark's difficulty levels. Finally, the results reveal a clear performance gap between model types, with leading closed-source models like Claude-3.5-Sonnet (74.5%) consistently outperforming the best open-source models (55.8%), highlighting the current advantages of proprietary systems on such specialized tasks.

## D.2 ANALYSIS OF FAILURE CASES

Despite recent advancements, large multimodal models (LMMs) still exhibit several fundamental failure modes when tasked with complex geometric reasoning. Through our analysis, we identify and categorize three primary types of error:

Semantic Grounding Errors: This class of error occurs when the model fails to correctly associate visual annotations with their corresponding geometric entities. For instance, a model might mis-attribute the numerical value of a specific angle to an adjacent, unlabeled angle. While the model may occasionally identify such a contradiction through subsequent deductive steps, this process is unreliable. A more robust solution involves enhancing the model's architecture to support iterative self-correction, allowing it to perform consistency checks across its reasoning chain and rectify initial grounding errors.

Figure 11: Failure cases of NeSyGeo-Test.

**Deficiencies in Latent Property Inference**: Models often overlook implicit geometric properties that are not explicitly annotated in the diagram, thereby increasing the perceived complexity of the problem. A common example is the failure to infer the congruence of sides in an isosceles triangle, which leads the model into a spurious exploration of the solution space through exhaustive, case-based analysis. This frequently results in wasted computational budget on incorrect deductive paths, sometimes causing premature termination due to exceeding context length limits. Future work should focus on improving the model's perceptual acuity and multi-step reasoning capabilities, enabling it to prune invalid branches of logic efficiently, even when faced with incomplete information.

**Vulnerability to Deductive Fallacies**: The reasoning process can be derailed by irrelevant visual information or distractors within the image, causing the model to commit logical errors. This often results in the generation of self-contradictory statements or conclusions that are tangential to the problem's solution path. To mitigate this, we propose the integration of a formal verification module. Such a neuro-symbolic validator would serve as a safeguard, programmatically ensuring that each step in the chain-of-thought is a logically sound consequence of the established premises and previously derived facts, thereby enforcing a more rigorous deductive process.

We present additional failure cases of Gemini-2.0-Flash in Figures 11 and 12, where the error types made by the model during the geometric reasoning task are highlighted with corresponding colors.

## D.3  FUTURE DIRECTION

Addressing these challenges points toward several key research directions for advancing multimodal geometric problem-solving. The path forward likely involves a tighter integration of perceptual modules with symbolic reasoning engines, moving toward more capable architectures. Furthermore, progress will depend on the development of architectures that explicitly support stateful, long-horizon reasoning, allowing them to maintain and update a coherent "mental model" of the geometric state. Finally, enabling models to achieve self-improvement through reinforcement learning loops, where rewards are provided by feedback from formal verifiers or successful problem completion, represents a crucial frontier for creating more robust and reliable geometric reasoning agents.

**Question:** Right triangle FYE. Quadrilateral YEXA is a parallelogram. Sector FEH. What is the area of sector FEH?

**Choices:** A. 15.4,  B. 23.9,  C. 20.7,  D. 28.3

1. Identify the radius: The radius of sector FEH is FE.
2. Find FE: Triangle FYE is a right triangle, so we can use the Pythagorean theorem to find FE:

$$FE = \sqrt{FY^2 + YE^2} = \sqrt{4^2 + 8^2} = \sqrt{16 + 64} = \sqrt{80} = 4\sqrt{5}$$

3. Find the angle of the sector: Angle YFE is given as 30 degrees. Angle EFH is 135 degrees. Therefore, $\angle EFH = 135°$.
4. Calculate the area of the sector: The area of a sector is given by the formula:

$$S = \left(\frac{135}{360}\right) \times \pi \times \left(4\sqrt{(5)}\right)^2$$

$$S = \left(\frac{3}{8}\right) \times \pi \times 80$$

$$S = 30 \times \pi$$

$$S \approx 30 \times 3.14159 \approx 94.2477$$

**Question:** GCFVW is a regular polygon. WGK is an isosceles triangle. What is the area of triangle WGK?

**Choices:** A. 28.5,  B. 30.7,  C. 33.1,  D. 31.3

The interior angle of a regular pentagon is given by $\frac{(n-2)\times 180}{n}$, where $n$ is the number of sides.

In this case, $n = 5$, so the interior angle is $\frac{(5-2)\times 180}{5} = \frac{3\times 180}{5} = \frac{540}{5} = 108°$.

Thus, $\angle VGF = 108°$.

We are given that $\angle WGC = 60°$.

Since $\angle WGC = 60°$, we have $GW = GC = 8.5$.

$\triangle WGK$ is an isosceles triangle. We know that $GW = GK$. We have $\angle WGC = 60°$.

Since GCFVW is a regular pentagon, $GC = GW = 8.5$.

We have $\angle GWK = \angle GKW$.

……

Figure 12: Failure cases of NeSyGeo-Test.

# E    CASE ANALYSIS

## E.1    COMPARISON WITH OTHER POPULAR GEOMETRY DATASETS

To facilitate comparison of dataset characteristics synthesized by our method and other popular approaches, we showcase a randomly selected example from NeSyGeo-CoT alongside each of the different approaches in Figure 13. Geometry-3K is a manually synthesized dataset, while the remaining approaches employ automatic generation techniques. To ensure a fair comparison, we standardize the text format by removing model-guiding prompts and appending options when present. Furthermore, we annotate each sample with image pixels and CoT word counts.

Compared to other datasets, our dataset features clear, human-aesthetically pleasing images, high-quality step-by-step reasoning chains, symbolic form meta-information enabling subsequent image augmentation and mutation, and well-distributed conditional information between images and text. Additional examples of our NeSyGeo-CoT dataset can be found in the Appendix 16.

| Dataset | Text | Image | Image Symbolic Form | Chain of Thought | Answer |
|---|---|---|---|---|---|
| NesyGeo-CoT | Triangle UFV. J is reflection of V over FU. Circle N is the circumcircle of triangle UFV. What is the diameter of circle N? "Choices": "A.7.5, B.8.0, C.8.5, D.9.0" | (675 x 790) | Triangle(U,F,V)=(4.5,4.5,12), Mirror(J,V,Line(F,U)), Cir_circle(N,Triangle(U,F,V)) | Step1: Triangle UFV is isosceles with UF = FV = 4.5 and ∠F = 120°.To find the diameter of the circumcircle (Circle N), we first need to find the length of UV.Step3: Using the Law of Cosines in triangle UFV… (145 words) | D |
| Geometry-3k | Triangle R S T cong triangle X Y Z. Find y. "choices": ["5","14","15","35"], | (864 x 434) | "Equals(LengthOf(Line(T, R)), x+21)", "Equals(MeasureOf(Angle(T, R, S)), 4y-10)", "Equals(MeasureOf(Angle(Z, X, Y)), 3y+5)", "Equals(LengthOf(Line(Z, X)), 2x-14)" | \ | C |
| G-LLaVA | In the given diagram, if angle 1 has a measure of 35.0 degrees, what is the measure of angle 2? choices": A: 55° B: 135° C: 145° D: 155° | (165 x 56) | \ | Since angle 1 + angle 2 = 180°, and angle 1 = 35°, therefore angle 2 = 145°. Therefore, option C is selected. (19words) | C |
| Geomverse | If the arc length of the ABC sector is 12.85, the area of the BDEF parallelogram is 108, the BFGH shape is a combination of a rectangle and a semi-circle, the length of the FG side is 6, the perimeter of the BFGH shape is 48, the area of the BJKD square is 121 and the angle DBF is vertical to CBA, compute the length of the BC side of the ABC sector. Assume $\pi=3.14$. Round computations to 2 decimal places. | (811 x 1280) | \ | The perimeter of the BFGH shape is 48 and the length of the FG side is 6, so $2 * OtherSide + 6 + \frac{6 * 3.14}{2} = 48$. So $2 * OtherSide = 48 - 6 - \frac{6 * 3.14}{2} = 48 - 6 - \frac{18.84}{2} = 48 - 6 - 9.42 = 32.58$. Therefore, the length of the BF side is $\frac{32.58}{2} = 16.29$. The area of the BJKD square is 121, so the length of the BD side is $\sqrt{121} = 11$… (188 words) | 20.4 |
| AutoGeo | Render a clear and concise description of a image about geometric shapes. | (1280 x 1280) | \ | \ | Rectangle ABCD.Point E is positioned in a way that line EB is perpendicular to point B.F lies on line segment CB. |
| MAVIS-Instruct | Side CD materializes as an equilateral triangle. DEF is identified as a sector. FEGH is a square. HGI is in the form of a sector. Please provide the length of arc HI in sector HGI. | (4434 x 1596) | \ | Given that AB is 27, rectangle has equal oppsite edges, so CD is also 27. Since CDE is a equilateral triangle…(88 words) | 18*π |
| R-CoT | Give reasoning steps and answers. In the diagram, there is a circle E with radius 2.1. What is the circumference of the circle E? | (514 x 404) | \ | Step1: The circumference of a circle is calculated using the formula C = 2\u03c0r. Step2: Substituting the given value, EM = 2.1, we get C = 2\u03c0*2.1 = 4.2\u03c0. The answer is 4.2\u03c0… (34 words) | 4.2 |

Figure 13: Comparison of NeSyGeo-CoT dataset with other Popular Geometry Datasets. Geometry-3K is a manually synthesized dataset, while the remaining approaches employ automatic generation techniques. Our dataset features clear, human-aesthetically pleasing images, high-quality step-by-step reasoning chains, symbolic form meta-information enabling subsequent image augmentation and mutation, and well-distributed conditional information between images and text.

## E.2 CASES OF SYMBOLIC SOLVER

While Section 3.3 qualitatively discussed the limitations of symbolic solvers, this section presents a granular statistical analysis and case study using the **GeoGen** (Pan et al., 2025) framework to empirically substantiate these claims. Building upon the four foundational task configurations defined in GeoGen, we stochastically generated a corpus of 600 samples. Subsequently, we employed the Qwen3-32B model, utilizing a one-shot prompting strategy, to transfigure the raw symbolic problem formulations and solution traces into coherent natural language narratives.

As shown in Figure 14 and 15, a rigorous analysis of the synthesized data reveals three critical pathologies inherent to the current symbolic approach:

1. **Inefficiency in Complex Reasoning:** In scenarios with high visual complexity, the symbolic solver often overlooks simple, global solutions. Instead, it tends to focus excessively on local logical steps, leading to unnecessarily long and complicated reasoning paths.

2. **Distributional Bias:** To quantify the gap between symbolic generation and real-world data, we calculated the Jensen-Shannon (JS) Divergence (Lin, 1991) between the distributions of 600 stochastically synthesized samples and the real-world ground truth provided by GeoGen. As shown in Table 12, the symbolic method exhibits substantial divergence scores. We attribute this significant distributional misalignment to the rigid nature of rule-based systems, which lack the flexibility and semantic diversity found in human-generated data.

3. **Logical Errors from Solver Defects:** Internal flaws in the solver's implementation can occasionally lead to incorrect logical derivations. These errors propagate through the reasoning chain, resulting in factual hallucinations in the final output.

Table 12: Quantitative Analysis of Distributional Shift. We report the Jensen-Shannon (JS) Divergence between the distributions of the symbolic synthesis (GeoGen) and the real-world ground truth.

| Method | Predicate JS | Theorem JS |
| --- | --- | --- |
| GeoGen | 0.8337 | 0.5329 |

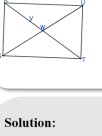

**Question:** Given $\angle URW = 34$, find the measure of $\angle SUR$. In the figure, RSTU is a rectangle, V is the centroid of $\triangle RSU$, V is on the line RW, S, W, U lie on the same line.

**Solution:**

1. <because> quadrilateral RSTU is a rectangle, <therefore> $UR \perp SR$, $\angle URS = 90$.
2. <by> right_triangle_judgment_angle, <because> $\angle URS = 90$ from step 1, <therefore> $\triangle$ URS has a right angle URS.
3. <by> centroid_of_triangle_property_intersection, <because> R, V, W are in one line, W lie on line SU, V is the centroid of $\triangle$ RSU from given condition, <therefore> RW is the median of $\triangle$ RSU.
4. <by> right_triangle_property_length_of_median, <because> $\triangle$ URS has a right angle URS from step 2, RW is the median of $\triangle$ RSU from step 3, <therefore> $UW = RW$.
5. <by> isosceles_triangle_judgment_line_equal, <because> $UW = RW$ from step 4, <therefore> $\triangle$ WUR is isosceles; $\angle WUR = \angle URW$.
6. Solve equations: <because>: $\angle WUR = \angle SUR$. $\angle WUR = \angle URW$. <therefore> $\angle SUR = \angle URW$. <because> $\angle$ URW = 34°, <therefore> $\angle$ SUR = 34°.

*Inefficiency in Complex Reasoning*

**Question:** Based on the figure provided, find the measure of $\frown$ DC. In the figure, quadrilateral ABCD is a square, EF is the midsegmet of quadrilateral ABCD, G is the circumcenter of quadrilateral ACDH, A, E, B are collinear, D, F, C are collinear, H, A, C, D are on circle G, $AE = EB$, $CF = FD$.

**Solution:**

1. <by> square_property_symmetry, <because> quadrilateral ABCD is a square from given condition, <therefore> ACD is an isosceles right triangle; DAC is a right triangle; $DA \perp CA$; $\angle DAC = 90$.
2. <by> arc_property_circumference_angle_internal, <because> D, A, C lie on the circle G from given condition, <therefore> $MAR\_GDC/2 + \angle DAC = 180$.
3. Solve equations: <because>: $MAR\_GDC/2 + \angle DAC = 180$. $\angle DAC = 90$. <therefore> $MAR\_GDC/2 = 90$. <therefore> measure of $\frown$ DC = 180.

*Logical Errors from Solver Defects*

Figure 14: Cases of symbolic reasoner.

### E.3 MORE EXAMPLES OF OUR DATASETS

We present more examples from the **NeSyGeo-CoT** dataset in Figure 16. Our bidirectional conversion engine can generate high-quality visual images from a symbolic form based on our Geo-DSL language. To further enhance image diversity, the engine introduces variability by randomly selecting values for the unit length and applying random rotations to the generated diagrams during creation. While other visual attributes, such as element and background colours, could also be randomized, they were set to default values in our current synthesis process. Our symbolic language helps identify parts of the image, and our conversion process ensures the images are geometrically correct.

### E.4 CASES OF REASONER AND VERIFIER

To provide a transparent illustration of our methodology, we present a concrete case study in Figure 17. This example details the step-by-step reverse search conducted by the Reasoner and the subsequent forward reasoning performed by the Verifier. Furthermore, to highlight the effectiveness of our

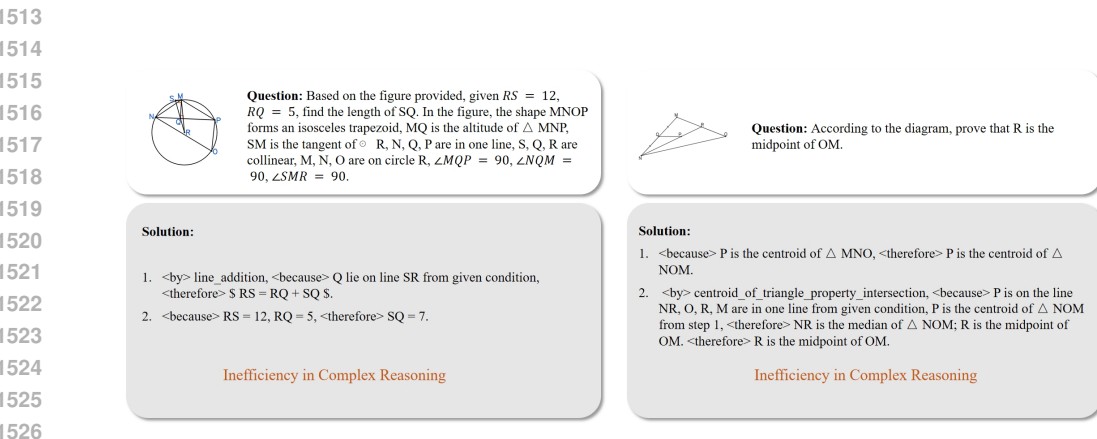

Figure 15: Cases of symbolic reasoner.

validation mechanism, we also showcase an instance in Figure 18that was filtered out due to an inconsistency detected during the verification phase.

## F    USE OF LLMS

Within the Reasoner and Verifier modules of NeSyGeo, we employ LLMs to perform deductive geometric reasoning. This is actualized through a two-stage generation process, comprising a step-by-step reverse search and a reliable forward validation. The formidable reasoning capabilities and extensive pre-trained knowledge inherent in LLMs facilitate the efficient expansion of reasoning states, while simultaneously ensuring the fidelity and diversity of the resulting reasoning paths.

All technical contributions, including the design of experiments, comparison of MLLMs (e.g., Qwen and GPT-4o), and interpretation of results, were fully conceived and conducted by the authors. The authors take full responsibility for the correctness, originality, and integrity of all scientific content presented in the paper.

## G    FUTURE WORK

We intend to extend NeSyGeo to other multimodal domains, such as analytical geometry and visual question answering. This extensibility will be achieved by defining new domain-specific languages, corresponding synthesis rules within the symbolic space, and tailored conversion engines. Furthermore, we plan to develop an automated symbolic solver capable of conducting search and validation directly within the symbolic space. This would remove reliance on LLMs, potentially reducing generation costs and ensuring complete correctness of the datasets.

## H    DETAILS OF DOMAIN-SPECIFIC LANGUAGES

### H.1    DEFINITION OF GEO-DSL

Geo-DSL adopts an entity-attributes-relations framework to comprehensively define geometric elements in plane geometry, systematically encompassing 13 types of points, 7 types of lines, 3 types of angles, and 14 types of shapes. By leveraging rigorous definition rules, the language uniquely

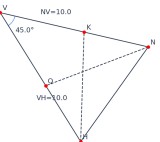

**Symbolic Form:** Iso_triangle(A,B,C)=(3,40), Cir_circle(O,Triangle(A,B,C))

**Element Types:** Triangle,Circle
**Difficulty Level:** 3

**Caption** : There is a isosceles triangle ABC, O is the circumcircle of ABC

**Question:** What is the measure of angle OCB?

**Choices:** A.60° B.40° C.20° D.10°

**Reasoning steps:** Step 1: Since angle ABC = 40° and BA = BC, angle ACB = (180° - 40°) / 2 = 70°. Step 2: The central angle subtended by the same chord is twice the inscribed angle, so angle AOC = 2 * 40° = 80°. Step 3: Since OA = OC = the radius of the circle, angle OCA = (180° - 80°) / 2 = 50°. Step 4: Therefore, angle OCB = 70° - 50° = 20°.

**Answer:** C.

**Symbolic Form:** Triangle(N,V,H)=(10.0,10.0,45), Angle_bisector(Q,Angle(H,N,V),Line(V,H)), Angle_bisector(K,Angle(V,H,N),Line(N,V))

**Element Types:** Triangle, Angle_bisector
**Difficulty Level:** 4

**Caption** : Consider triangle NVH. Angle HNV is bisected by ray Q. Line K divides angle VHN into two equal parts

**Question:** What is the length of side KN?

**Choices:** A.4.3 B.5.3 C.6.3 D.7.3

**Reasoning steps:** Step1: Given triangle NVH with VH = NV = 10 and angle V = 45°. Step2: The Angle Bisector Theorem states VK/KN = VH/HN. HN can be found using the Law of Cosines in triangle NVH: HN² = NV² + VH² - 2*NV*VH*cos(V) = 10² + 10² - 2*10*10*cos(45°) ≈ 200 - 141.42 ≈ 58.58, so HN ≈ 7.7. Step3: According to the Angle Bisector Theorem, VK/KN = VH/HN ≈ 10/7.7 ≈ 1.3. Step4: Let VK = 1.3x and KN = x. Then VK + KN = VN = 10, so 1.3x + x = 10 → 2.3x = 10 → x ≈ 4.3. Therefore, KN ≈ 4.3.

**Answer:** A.

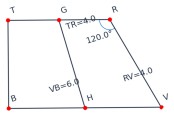

**Symbolic Form:** Trapezoid(T,R,V,B)=(4,4,6,120) IsMidpointOf(G,Line(R,T)) IsMidpointOf(H,Line(V,B)) Line(G,H)

**Element Types** : Trapezoid, Midpoint
**Difficulty Level:** 5

**Caption** :A trapezoid with vertices TRVB.G divides line segment RT into two equal parts.Point H is the midpoint of line segment VB. Line GH.

**Question:** What is the area of shape TGHB?

**Choices:** A.11.3 B.13.5 C.5.7 D.8.7

**Reasoning steps:** Step1: Draw trapezoid TRVB with TR∥VB, TR=4,VB=6, RV=4, ∠TRV=120°. First, Calculate height (h):Drop a perpendicular from R to VB at P. In ΔRVP: ∠RVP=60°, RV=4 → h=4×sin60°=2√3≈3.464. Step2: Find midpoints:G is midpoint of RT → GT = 2. H is midpoint of VB → VH=3. Step3: Calculate areas:Total trapezoid area = ½(4+6)×2√3=10√3≈17.32. Step4: ΔHRG area =½×2×2√3=2√3 ≈3.464.ΔVHR area =½×3×2√3=3√3≈5.196. Step5: Compute TGHB area: 17.32 - 3.464 - 5.196 ≈ 8.66 ≈ 8.7.

**Answer:** C.

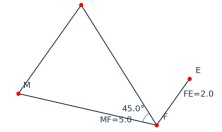

**Symbolic Form:** Iso_triangle(M,F,Q)=(5,45) Paral(Line(E,F),Line(M,Q),2)

**Element Types:** Triangle,
**Difficulty Level:** 2

**Caption** : Triangle MFQ is isosceles.We have line MQ parallel to line FE

**Question:** What is the measure of angle QFE?

**Choices:** A.45° B.67.5° C.73.5° D.53.5°

**Reasoning steps:** Step1: Triangle MFQ is isosceles with MF = FQ (given). Therefore, angle FMQ = angle MQF. angle FMQ + angle MQF = 180 - 45 = 135 degrees. Since angle FMQ = angle MQF, each angle is 135 / 2 = 67.5 degrees. Step2: Since MQ is parallel to FE and FQ is a transversal, angle MQF is equal to angle QFE by the alternate interior angles theorem, therefore angle QFE = angle MQF = 67.5 degrees.

**Answer:** B.

Figure 16: Examples of the **NeSyGeo-CoT** dataset. Each sample comprises a symbolic image definition based on our Geo-DSL language, a high-quality annotated image, a concise text caption, diverse Q&A pairs, and a detailed reasoning process step-by-step.

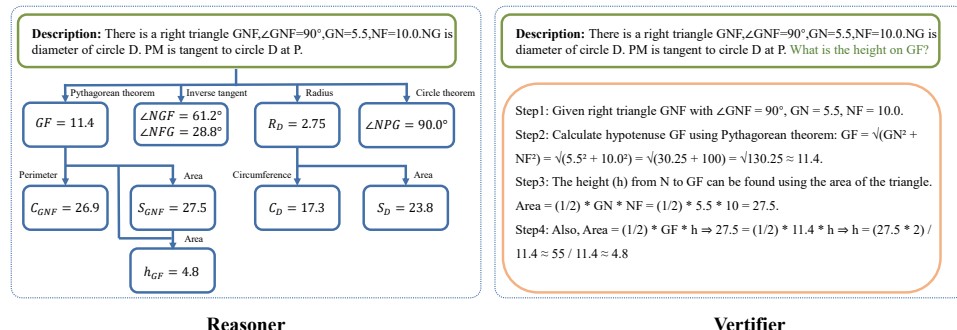

Figure 17: Case of reasoner and verifier.

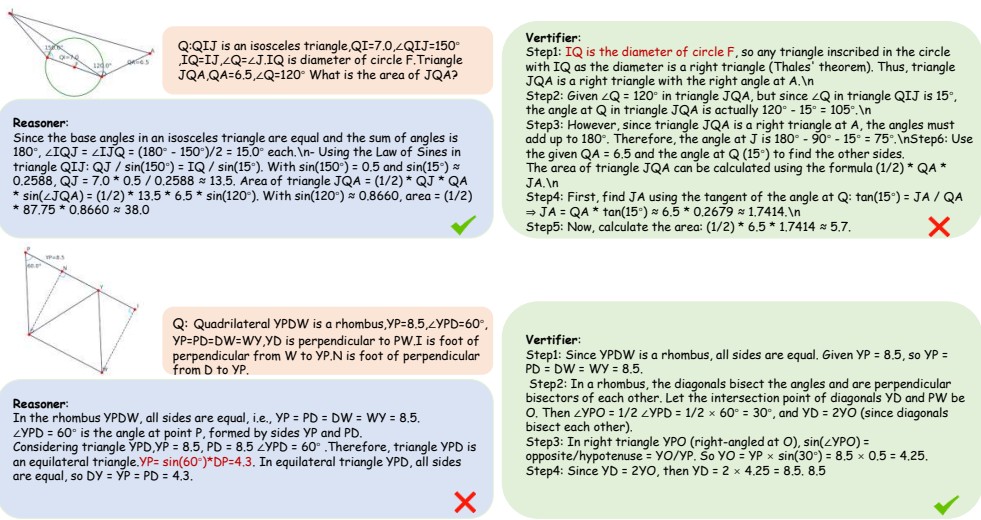

Figure 18: Cases filtered out due to an inconsistency detected during the verification phase.

specifies spatial elements and numerical attributes—such as lengths and angle measures—within a single concise statement. This design ensures precise geometric representations and significantly reduces symbolic processing complexity, thereby facilitating efficient parsing by our conversion engine and seamless integration with neural synthesis pipelines.

Representative examples of these symbolic statements and their natural language descriptions are illustrated in Figures 24, 21, 22, and 23.

We posit that this taxonomy covers all fundamental primitives of plane geometry. To empirically validate this expressivity, we successfully synthesized diagrams across three distinct tiers of complexity: (1) **basic elements**, grounded in the knowledge points from the MathBook Knowledge System of We-Math (Qiao et al., 2025); (2) **complex problems**, representative of standard high school curriculum difficulty; and (3) **advanced challenges**, corresponding to the level of IMO competitions. Visualizations of these synthesized instances are presented in Figure 19, demonstrating Geo-DSL as a robust and versatile solution for generating high-quality multimodal geometric reasoning data.

## H.2 Definition of Actions in Symbolic Sequence Generation

As illustrated in Figure 20, we enumerate all statements within the action space defined by our Geo-DSL. The content within square brackets denotes annotations for each statement.

As outlined in Algorithm 1, for each step, we first generate three lengths, $x$, $y$, and $z$, along with an angle $\alpha$, sampled from predefined ranges. Subsequently, based on the weight matrices $A$ and $I$, we

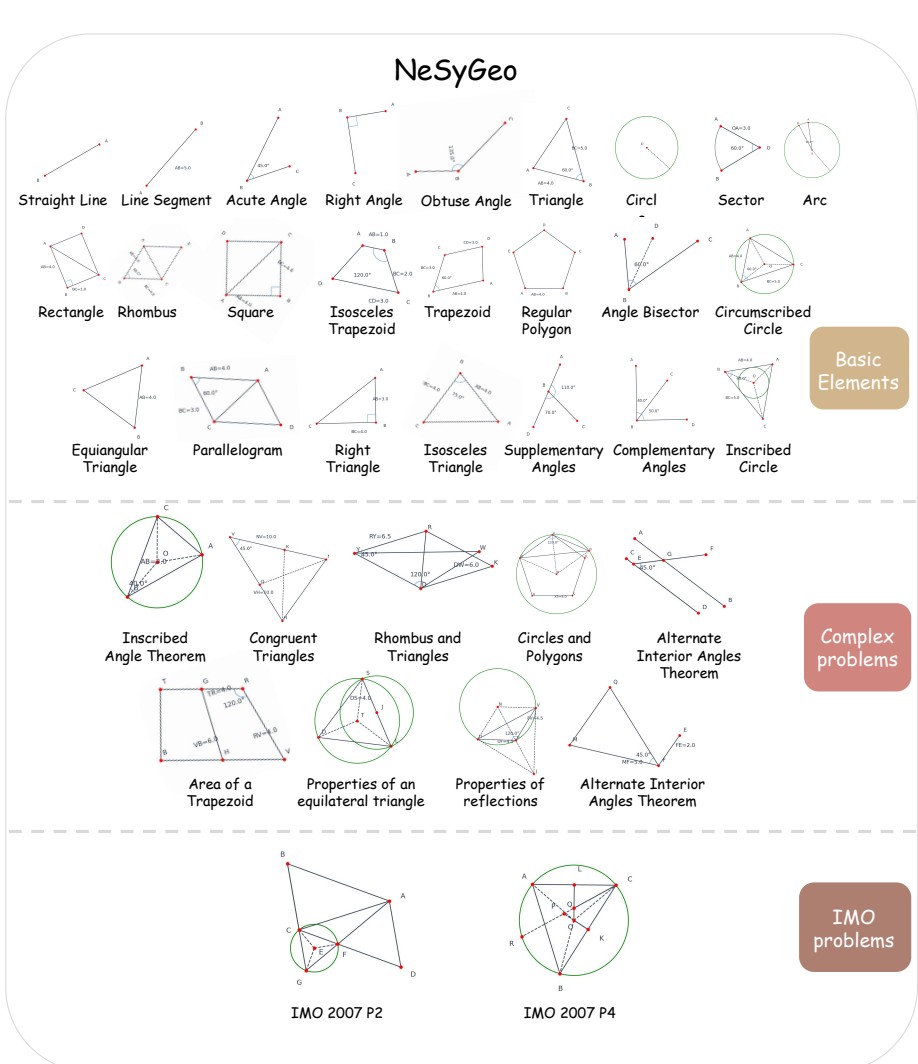

Figure 19: Examples of plane geometry diagrams generated using Geo-DSL.

determine the specific statement to be selected. The chosen statement, paired with its corresponding numerical values, is then appended to the Geo-DSL sequence. For different types of actions, we provide  a concrete example  for each, highlighted with a gray background to indicate the available action space when the respective element is selected.

**Line-based Actions**

Chosen element: Line(A,B)

Triangle(A,B,C)=(,x,α)

R_triangle(A,B,C)=( ,y)

Iso_triangle(A,B,C)=( ,α)

Ieq_triangle(A,B,C)=()

Parallelogram(A,B,C,D)=( ,x,α)

Cir_circle(O,Triangle(A,B,C))

Rectangle(A,B,C,D)=( ,x)

Rhombus(A,B,C,D)=(,α)

Square(A,B,C,D)=( )

Re_Polygon(A,B,C,D,E)=( )

Trapezoid(A,B,C,D)=( ,x,,α)

IsOnline(D,Line(A,B),x)

Mirror(X,C,Line(A,B))

IsMidpointOf(C,Line(A,B))

Intersection_cl(X,Circle(O),Line(A,B))

Intersection_ll(X,Line(A,B),Line(C,D))

Para (Line(C,D),Line(A,B),x)

Perp (Line(C,D),Line(A,B),x)

Angle_bisector(X,Angle(E,D,F),Line(A,B))

Sector(A,B,O)=( ,α)
[The first and the second point can exchange]

Chosen element: Arc(A,B)

Diameter(Line(E,F),Circle(O))

Arc(E,F)=(O,α)

Intersection_cl(X,Circle(O),Line(E,F))

Intersection_cc(X,Circle(O),Circle(P))

Sector(O,A,B)=(r,α)
[The first point may exist on the arc of circle.]
Ins_angle(A,B,C,Circle(O),α)
[The first point may exist on the arc of circle.]
Cen_angle(B,C,Circle(O),α)
[The first point may exist on the arc of circle.]
Tangent(Circle(O),Line(P,T),x)
[The first point of line may exist on the arc of circle.]

**Point-based Actions**

Chosen element: Point(A)

Foot(X,A,Line(B,C))

Circle(A)=(x)

Mirror(X,A,Line(B,C))

In_circle(O,Triangle(A,B,C))

Cir_circle(O,Triangle(A,B,C))

Tangent(Circle(O),Line(B,A),x)

Free(B)

Line(A,D)=(x)
[Point D undefined before]
Line(A,D)=( )
[Point D must exist]
Foot(X,A,Line(B,C))
[Randomly select a vertex of a triangle and its opposite side.]

**Shape-based Actions**

Chosen element: Triangle(A,B,C)

IsCentroidOf(Q,Triangle(A,B,C))

IsCenterOf(Q,Triangle(A,B,C))

IsOrthoOf(Q,Triangle(A,B,C))

IsIncenterOf(Q,Shape(A,B,C))

In_circle(O,Triangle(A,B,C))

Cir_circle(O,Triangle(A,B,C))

Foot(X,A,Line(B,C))
[Randomly select a vertex of a triangle and its opposite side.]
Intersection_ll(X,Line(A,B),Line(C,E))
[Randomly select a vertex of a triangle and its opposite side.]
Para (Line(E,C),Line(A,B),x)
[Randomly select a vertex of a triangle and its opposite side.]
Perp(Line(E,C),Line(A,B),x)
[Randomly select a vertex of a triangle and its opposite side.]
Angle_bisector(X,Angle(A,B,C),Line(A,C))
[Randomly select an angle of a triangle and its opposite side.]
Mirror(X,C,Line(A,B))
[Randomly select a vertex of a triangle and its opposite side.]

Chosen element: Circle(O)
Diameter(Line(E,F),Circle(O))

Arc(E,F)=(O,α)

Intersection_cl(X,Circle(O),Line(E,F))

Intersection_cc(X,Circle(O),Circle(P))

Sector(O,A,B)=(r,α)
[The first point may exist on the arc of circle.]
Ins_angle(A,B,C,Circle(O),α)
[The first point may exist on the arc of circle.]
Cen_angle(B,C,Circle(O),α)
[The first point may exist on the arc of circle.]
Tangent(Circle(O),Line(P,T),x)
[The first point of line may exist on the arc of circle.]

Chosen element:Parallelogram(A,B,C,D)
IsIncenterOf(O,Shape(A,B,C,D))

Line(A,C)
[Randomly select two groups of non-adjacent vertices.]
Intersection_ll(X,Line(A,B),Line(C,D))
[Randomly select two groups of non-adjacent vertices.]
Foot(X,C,Line(A,B))
[Randomly select a vertex of a polygon and a side that exclude that vertex.]
Intersection_ll(X,Line(D,E),Line(A,B))
[Randomly select a vertex of a polygon and a side that exclude that vertex.]
Para(Line(E,D),Line(A,B),x)
[Randomly select a vertex of a polygon and a side that exclude that vertex.]
Perp(Line(E,D),Line(A,B),x)
[Randomly select a vertex of a polygon and a side that exclude that vertex.]
Mirror(X,C,Line(A,B))
[Randomly select a vertex of a polygon and a side that exclude that vertex.]

**Angle-based Actions**

Chosen element: Angle(A,B,C)

Line(A,C)=()

Angle_bisector(D,Angle(A,B,C),x)

Angle_bisector(D,Angle(A,B,C),Line(E,F))

Parallelogram(A,B,C,D)=( , , )

Trapezoid(A,B,C,D)=( , ,z, )

Triangle(A,B,C)=( , , )

Figure 20: Detailed Actions in Symbolic Spaces. Actions can be categorized into four parts based on the type of selected geometric element: line-based, point-based, shape-based, and angle-based.

| Geo-DSL Language | Natural Language | Notes |
|---|---|---|
| Free(A) | A is a random point | |
| IsOnline(A,Line(B,C),x) | A is point on Ray BC, BA=x | Line BC must be predefined, value x can be omitted |
| IsOnarc(A,Circle(O)) | A is point on arc of circle O | Circle O must be predefined |
| Mirror(X,C,Line(A,B)) | X is reflection of C over AB | Line AB and point C must be predefined |
| Foot(X,C,Line(A,B)) | X is foot of perpendicular from C to AB | Line AB and point C must be predefined |
| IsMidpointOf(C,Line(A,B)) | C is midpoint of AB | Line AB and point C must be predefined |
| IsCentroidOf(D,Triangle(A,B,C)) | D is centroid of triangle ABC | Triangle ABC must be predefined |
| IsCenterOf(D,Triangle(A,B,C)) | D is incenter of triangle ABC | Triangle ABC must be predefined |
| IsOrthoOf(D,Triangle(A,B,C)) | D is orthocenter of triangle ABC | Triangle ABC must be predefined |
| IsIncenterOf(O,Shape(A,B,C,D)) | O is center of shape ABCD | Shape must be predefined and has more than two points |
| Intersection_cl(X,Circle(O),Line(A,B)) | X is intersection of circle O and AB near A | Circle O and line AB must be predefined |
| Intersection_ll(X,Line(A,B),Line(C,D)) | X is intersection of AB and CD | Line AB and line CD must be predefined |
| Intersection_cc(X,Circle(O),Circle(P)) | X is intersection of circles O and P | Circle O and circle P must be predefined |

Figure 21: Geo-DSL definitions of point.

| Geo-DSL Language | Natural Language | Notes |
|---|---|---|
| Angle(A,B,C)=(α) | Angle ABC = α° | Value α can be omitted |
| Ins_angle(A,B,C,Circle(O),α) | Angle ABC is inscribed angle of circle O, equals to α° | Circle O must be predefined, value α can be omitted |
| Cen_angle(B,C,Circle(O),α) | Angle BOC is central angle of circle O, equals to α° | Circle O must be predefined, value α can be omitted |

Figure 22: Geo-DSL definitions of angle.

| Geo-DSL Language | Natural Language | Notes |
|---|---|---|
| Triangle(A,B,C)=(x,y,α) | Triangle ABC has AB=x, BC=y, angle B=α° | Value x, y or α can be omitted |
| R_triangle(A,B,C)=(x,y) | Right triangle ABC has angle B=90°, AB=x, BC=y | Value x or y can be omitted |
| Iso_triangle(A,B,C)=(x,α) | Isosceles triangle ABC with side AB=x, vertex angle B=α° | Value x or α can be omitted |
| Ieq_triangle(A,B,C)=(x) | Equilateral triangle ABC has AB=x | Value x can be omitted |
| Parallelogram(A,B,C,D)=(x,y,α) | Parallelogram ABCD has AB=x, BC=y, angle B=α° | Value x, y or α can be omitted |
| Rectangle(A,B,C,D)=(x,y) | Rectangle ABCD has AB=x, BC=y | Value x or y can be omitted |
| Rhombus(A,B,C,D)=(x,α) | Rhombus ABCD has AB=x, angle A=α° | Value x or α can be omitted |
| Square(A,B,C,D)=(x) | Square ABCD has AB=x | Value x can be omitted |
| Re_Polygon(A,B,C,D,E)=(x) | Regular polygon ABCDE has AB=x | Value x can be omitted |
| Trapezoid(A,B,C,D)=(x,y,z,α) | Trapezoid ABCD has AB=x, BC=y, CD=z, angle ABC=α° | Value x, y, z or α can be omitted |
| Circle(O)=(r) | Circle O has radius=r | Value r can be omitted |
| Sector(O,A,B)=(r,α) | Sector OAB with O is the center, has radius=r, central angle=α° | Value r or α can be omitted |
| In_circle(O,Triangle(A,B,C)) | Circle O is incircle of triangle ABC | Triangle ABC must be predefined |
| Cir_circle(O,Triangle(A,B,C)) | Circle O is circumcircle of triangle ABC | Triangle ABC must be predefined |

Figure 23: Geo-DSL definitions of shape.

| Geo-DSL Language | Natural Language | Notes |
|---|---|---|
| Line(A,B)=(x) | Line AB = x | Value x can be omitted |
| Arc(A,B)=(O,α) | Arc AB on circle O is α ° | Circle O must be predefined, value α can be omitted |
| Para(Line(A,B),Line(C,D),x) | Line AB is parallel to CD, AB=x | Line CD must be predefined, value x can be omitted |
| Perp(Line(A,B),Line(C,D),x) | Line AB is perpendicular to CD, AB=x | Line CD must be predefined, value x can be omitted |
| Tangent(Circle(O),Line(P,T),x) | Line PT is tangent to circle O at P, PT=x | Circle O mest be predefined, value x can be omitted |
| Angle_bisector(X,Angle(A,B,C),x) | Line BX is bisector of angle ABC, BX=x | Angle ABC must be predefined, value x can be omitted |
| Angle_bisector(X,Angle(A,B,C),Line(E,D)) | Angle ABC bisector intersects DE at X | Angle ABC and Line ED must be predefined, value x can be omitted |

Figure 24: Geo-DSL definitions of line.

