# OpenReview forum: "NeSyGeo: A Neuro-Symbolic Framework for Multimodal Geometric Reasoning Data Generation"
_ICLR.cc/2026/Conference — Submitted to ICLR 2026_

### Official Review · Reviewer_4nkp · 2025-10-28

**Soundness:** 2
**Presentation:** 3
**Contribution:** 2
**Rating:** 4
**Confidence:** 5

**Summary:**

The paper introduces NeSyGeo, a geometry data generation framework that leverages a novel domain-specific language (Geo-DSL) to automatically produce large-scale, high-quality multimodal datasets. The framework encodes geometric entities, relationships, and constraints symbolically, then generates corresponding images and natural language descriptions through an automated pipeline. By utilizing LLMs for chain-of-thought (CoT) reasoning generation and LMMs for cross-validation, NeSyGeo produces two key datasets—NeSyGeo-CoT and NeSyGeo-Caption—that demonstrate significant performance improvements for multimodal large language models on geometric reasoning tasks.

**Strengths:**

1.	Comprehensive experimental validation: The authors provide extensive experiments including human evaluation (40 experts), ablation studies, contamination analysis, and comparisons across multiple benchmarks.
2.	Through RL and SFT, models trained on NeSyGeo data outperform those using standard and other synthetic datasets.

**Weaknesses:**

1.	The individual components (symbolic generation, LLM-based CoT distillation, template-based translation) lack novelty. Compared to AlphaGeometry, TrustGeoGen, and recent symbolic-neural integration works, the advantages are not sufficiently distinct.
2.	The framework heavily relies on DeepSeek-R1 and DeepSeek-V3 for critical reasoning and verification steps. This distillation-like approach may limit both the difficulty ceiling of the generated data and the performance upper bound of trained models.
3.	Although the authors claim Geo-DSL achieves "descriptive completeness" with 37 core statements, the paper lacks formal proof or systematic analysis of its coverage over plane geometry.

**Questions:**

1.	What are the computational requirements for generating 100k samples? How does this compare to other synthesis methods in terms of time and resource consumption?

---

> ### Author Response · Authors · 2025-11-20
> **Response to Reviewer 4nkp (Part 1)**
>
> > **Q1:Compared to AlphaGeometry, TrustGeoGen, and recent symbolic-neural integration works, the advantages are not sufficiently distinct.**
>
> We value your critique regarding the novelty of our components. Here's the main limitation of existing works and the clearification of our framework.
> - Symbolic Representation Constraints:
>   - AlphaGeometry is designed exclusively for theorem proving. Its symbolic space **lacks representations of numerical values of attributes**,which cannot provide verifiable numerical answers required for RLHF etc.
>   - TrustGeoGen relies on a "Base Scene Pool," limiting its symbolic space to **combinations of pre-existing template**,which restricts the structural diversity of the generated data.
> - CoT Generation Constraints:
>
>    Both works rely **on rigid, rule-based symbolic solvers for CoT generation**, which present some limitations: **Lack of Diversity** : Rigid solvers produce monotonous solution paths, failing to capture the diverse linguistic ways numerical reasoning can be expressed,which increases the risk of model overfitting. **Distributional Bias**: The inherent rigidity of symbolic solvers leads to significant distributional misalignment with real-world data. **Computational Overhead**: Each search step requires traversing the entire theorem space, resulting in high time complexity. **Poor Generalizability**: Constructing such solvers is complex and domain-specific, making them difficult to adapt to new domains.  **We have provided a more detailed analysis of these symbolic solver limitations in Appendix E.2 of the revised manuscript.**
>
> To overcome these systemic issues, NeSyGeo introduces specific innovations:
>
> - Geo-DSL: We propose **a concise yet comprehensive DSL that with Entity–Attribute–Relation Framework**,which ensures the completeness and diversity of the symbolic space.
> - Reasoner: We employ**Reverse BFS with Iterative Expansion guided by LLM priors**, which  leverages world knowledge to accelerate search and introduce linguistic diversity and mitigates the hallucination propensity common in pure LLM generation.
>
> Furthermore, other modules also **address specific challenges**: the Generator employs a Step-Action Augmentation strategy to guarantee symbolic sampling diversity, while the Painter utilizes dynamic annotation strategies and the Information Orthogonality principle to eliminate textual shortcuts.
>
> > **Q2:The framework heavily relies on DeepSeek-R1 and DeepSeek-V3 for critical reasoning and verification steps. This distillation-like approach may limit both the difficulty ceiling of the generated data and the performance upper bound of trained models.**
>
> Thanks for pointing it out!
>
> For Visual Complexity, our iterative Generator can synthesize diagrams of arbitrary complexity simply by increasing the sequence steps ($N$), **ensuring no inherent visual upper bound.**
>
> As for reasoning paths, **we respectfully disagree that using DeepSeek-R1/V3 limits the difficulty ceiling or performance upper bound. Our Reverse Step-wise BFS strategy utilizes LLMs not as end-to-end generators, but as search operators within a constrained symbolic graph**. In our pipeline, the Reasoner does not "hallucinate" a problem from scratch,as the ablation study shown in Table 1. It performs the Search on a valid symbolic state $S_t$ to find a logical predecessor. **The difficulty is determined by the depth of this search and the complexity of the geometric configuration, not merely the model's internal knowledge.**
> Consequently, the generated data remains sufficiently challenging to drive significant improvements in current models.
>
> This is empirically validated by our new experiments, where we fine-tuned Qwen2.5-VL-7B-Instruct using varying proportions (30%, 50%, 80%, and 100%) of our dataset combined with Geo170K (117k real-world samples), **observing steady performance gains on the GeoQA benchmark.**
>
> | Setting      | Base  | Geo170k | +30% Ours | +50% Ours | +80% Ours | +100% Ours |
> | ------------ | ----- | ------- | --------- | --------- | --------- | ---------- |
> | **Accuracy** | 69.4% | 72.3%   | 72.8%     | 74.3%     | 76.0%     | 77.2%      |

---

> ### Author Response · Authors · 2025-11-20
> **Response to Reviewer 4nkp (Part 2)**
>
> > **Q3:Although the authors claim Geo-DSL achieves "descriptive completeness" with 37 core statements, the paper lacks formal proof or systematic analysis of its coverage over plane geometry.**
>
> Thanks for your question!  We appreciate the opportunity to clarify the claim of "descriptive completeness." **We want to clear that this refers to the representational capacity for Plane Geometry as defined in standard educational curricula (e.g., from basic elements to Olympiad level), rather than an exhaustive axiomatic formalization of all Hilbert geometry**.
>
> Our DSL employs an entity-relation-constraint framework, utilizing well-defined rules to uniquely specify 13 types of points, 7 types of lines, 3 types of angles, and 14 types of shapes. **We posit that this set encompasses all fundamental primitives of plane geometry, while corresponding extension statements ensure expressivity across the entire geometric space.**
>
> To further validate this coverage, we successfully synthesized diagrams across three distinct tiers: **(1) basic elements grounded in the knowledge points from the MathBook Knowledge System of We-Math [1]; (2) complex problems representative of high school difficulty; and (3) advanced challenges at the IMO competition level**. These examples are visualized in Figure 19 of the revised  script.
>
> > **Q4:What are the computational requirements for generating 100k samples? How does this compare to other synthesis methods in terms of time and resource consumption?**
>
>   **We utilized an AMD Ryzen 9 7940H processor, completing the synthesis of the 100k dataset in approximately two days.**
>
> Given that most existing data synthesis methods have not released their source code, we selected the open-source framework GeoGen[1]  to conduct a fair baseline comparison. To ensure statistical reliability, we synthesized 5,000 images and 600 QA pairs for each method under consistent thread settings to calculate the average time and resource consumption. Notably, to ensure parity in multimodal generation tasks, we integrated an additional Qwen3-32B inference step into the final stage of the GeoGen pipeline for symbolic-to-natural language translation. The results are detailed below.
>
> |        | s/Image | s/QA Pair  | Tokens/QA Pair |
> | ------ | ------ | ------- | ------------ |
> | GeoGEN | 0.244s | 10.852s | 1323.94      |
> | Ours   | 0.208s | 7.203s  | 1150.66      |
>
>  **Considering the one-off nature of data synthesis, the cost disparity between our method and GeoGen is negligible.**
>
> ### References
>
> [1] Y. Pan et al., "Enhancing the Geometric Problem-Solving Ability of Multimodal LLMs via Symbolic-Neural Integration," ACM MM, 2025.

---

> ### Comment · Reviewer_4nkp · 2025-11-26
> **Official Comment by Reviewer 4nkp**
>
> Thank you for the additional explanation; however, it still does not clearly answer how the proposed framework provides distinct advantages over AlphaGeometry, TrustGeoGen, and other symbolic–neural approaches for the stated goal. I tend to maintain my original rating.

---

> > ### Author Response · Authors · 2025-11-26
> > **Response to Reviewer 4nkp**
> >
> > We thank the reviewer for the response. We understand the concern regarding overlap with existing neuro-symbolic frameworks. However, we believe that **NeSyGeo addresses specific, critical gaps in multimodal data generation that frameworks like AlphaGeometry and TrustGeoGen do not solve.**
> >
> > Our framework provides four distinct advantages:
> >
> > **1. Comprehensive Symbolic Language with Numerical Capabilities**
> >
> > - **Limitation of AlphaGeometry:** Designed exclusively for formal logic, its symbolic space lacks representations for **metric attributes** (e.g., specific angle degrees, lengths). This renders it fundamentally unsuitable for generating the numerical VQA data required for standard MLLM benchmarks.
> > - **Limitation of TrustGeoGen:** Instead of a rigorous atomic language, TrustGeoGen relies on a "Base Scene Pool" and **fixed topological templates**. This dependence restricts its symbolic representation to variations of pre-existing patterns, limiting its expressiveness.
> > - **Advantage of NeSyGeo:** We propose **Geo-DSL**, a **concise yet comprehensive language** explicitly architected around the **Entity-Attribute-Relation** framework. It supports precise **numerical values and calculations**, enabling the generation of calculation-heavy Q&A pairs. Furthermore, this rigorous symbolic foundation serves as the deterministic basis that enables accurate parsing by our Generator and Painter modules.
> >
> > **2. High-Quality and Linguistically Diverse Reasoning Paths**
> >
> > - **Limitation of  Symbolic Solvers:** Methods like AlphaGeometry and TrustGeoGen rely heavily on deterministic symbolic solvers, which often produce reasoning paths that are logically valid but "machine-like"—**overly verbose or unnatural compared to human reasoning**. Specifically, in Appendix E.2, we conducted a Jensen-Shannon Divergence analysis showing that **solver-based methods exhibit a significant distributional shift from real-world data**.  Moreover, Symbolic solvers are not always effective: they **suffer from inefficiency in complex reasoning, distributional bias, and poor generalizability**, as discussed in Section 3.3 and Appendix E.2.
> >
> > - **Advantage of NeSyGeo:** By integrating LLMs directly into the **Reasoning** phase via an **LLM-guided Reverse Search and Forward Validation** pipeline, NeSyGeo generates "soft," linguistically diverse reasoning paths. This **mimics human intuition while maintaining logical rigor, preventing the model from overfitting to rigid solver patterns and improving distillation efficiency.**
> >
> > **3. Unbounded Generative Diversity**
> >
> > - **Limitation of TrustGeoGen:** TrustGeoGen employs an incremental generation approach that relies on expanding a **"Base Scene Pool."** This dependence inherently restricts structural diversity to variations of pre-defined topological templates.
> > - **Advantage of NeSyGeo:** Our **Generator** employs a **Step-Action Augmentation** strategy that synthesizes geometries **from scratch** using **atomic symbolic actions** defined in Geo-DSL. By operating at the atomic level rather than the scene-template level, NeSyGeo achieves **unbounded topological diversity**, creating novel geometric configurations that template-based methods cannot produce.
> >
> > **4. Visual Grounding via Information Orthogonality**
> >
> > - **Limitation of Existing Approaches:** Most symbolic-neural or solver-based approaches suffer from **Information Redundancy**. They typically include all geometric conditions in the text prompt, allowing MLLMs to solve the problem **using text-only shortcuts without looking at the image**.
> > - **Advantage of NeSyGeo:** We enforce **Information Orthogonality**. Our Painter module embeds numerical values only in the image , while the Translator actively prunes these values from the text. This forces the MLLM to **perform true cross-modal reasoning rather than text-only reasoning.** Our ablation studies on "Vision Only" benchmarks prove that this specific design yields superior visual perception gains compared to redundancy-heavy datasets.
> >
> > Therefore, we believe our proposal makes solid contributions compared to existing methods, **making NeSyGeo a uniquely effective framework for the specific goal of enhancing MLLM visual geometric reasoning**, as evidenced by the 15.8% improvement on MathVision and 7.3% on GeoQA.
> >
> > Directly addressing the bottlenecks of existing methods is a non-trivial task. Our approach effectively bridges this gap, **positioning NeSyGeo as a uniquely effective framework for the specific goal of enhancing MLLM visual geometric reasoning**. This efficacy is confirmed by a 15.8% margin on MathVision and 7.3% on GeoQA.

---

### Official Review · Reviewer_ybzn · 2025-10-30

**Soundness:** 3
**Presentation:** 4
**Contribution:** 3
**Rating:** 6
**Confidence:** 3

**Summary:**

The paper proposes a novel neuro-symbolic framework called NeSyGeo to address the shortcomings of MLLMs in visual reasoning for geometry problems. More specifically, authors made NeSyGeoCoT and NeSyGeo-Caption datasets, containing 100k samples, and also release a new benchmark NeSyGeo-Test (~2.7k samples) to the community. Post-training using RL and SFT via the dataset shows performance boost on multiple benchmarks.

**Strengths:**

1. The core problem identified and solved by authors is quite interesting as models can ignore images while solving geometry questions in the existing datasets because of their text-image redundancy.

2. The framework combines the strength of symbolic Geo-DSL and flexibility of neural models and the Reasoner-Verifier paradigm helps create diverse with correct reasoning paths.

3. Generated data is diverse and has a high quality and training is efficient-- via only 4k samples and two RL epocs, the 4B model outperforms 8B model on multiple datasets. Benchmark is also novel and can be quite helpful for the community.

4. Extensive experiments, failure analysis and detailed implementations make the paper reliable and the framework's reproducibility easier.

**Weaknesses:**

1. My only concern is while the method is quite limited to plane geometry only, it's complexity (multi-stage, API calls, etc.) is high and it heavily depends on powerful large-scale teacher models (e.g., DeepSeek and GPT).

**Questions:**

Based on the above comments, I have a few questions:

1. How can authors overcome the usability limitation of the method to be able to extend it beyond plane geometry? Detailed explanation would shed light on the future iterations.

2. What happens if we swap the teacher models with smaller models? Does the performance drop drastically? Is there a minimum capability threshold that a model should have to be considered safe? I would recommend adding some experiments to show that.

3. RE information orthogonality in 3.4, what is the optimal balance between text and image modalities? I couldn't find any information RE numerical data in the text, so my bad if it's already there, but how performance would change if we include such data in text?

---

> ### Author Response · Authors · 2025-11-20
> **Response to Reviewer ybzn (Part 1)**
>
> > **Q1:My only concern is while the method is quite limited to plane geometry only, it's complexity (multi-stage, API calls, etc.) is high and it heavily depends on powerful large-scale teacher models (e.g., DeepSeek and GPT).**
>
> Thank you for this observation！ We consider the complexity of our framework **a necessary trade-off to guarantee exceptional data quality**.
>
> As highlighted in our Introduction, synthesizing multimodal geometric data presents significant challenges across **both visual and textual modalities**. A simplistic pipeline is **insufficient** to simultaneously ensure the diversity of the synthesis space, visual precision, information orthogonality between text and image, and the logical soundness of solution paths. **By adopting a neuro-symbolic approach to decouple these distinct tasks, our framework successfully achieves the synthesis of a high-fidelity, high-quality, and large-scale dataset**.
>
> We also benchmarked our method against the open-source GeoGen[1] framework. To evaluate resource consumption, we synthesized 5,000 images and 600 QA pairs for each method under consistent thread settings. To ensure parity in multimodal generation, we integrated an additional Qwen3-32B inference step into the GeoGen pipeline for symbolic-to-natural language translation. The comparative results are detailed below.
>
> |        | s/Image | s/QA Pair  | Tokens/QA Pair |
> | ------ | ------ | ------- | ------------ |
> | GeoGEN | 0.244s | 10.852s | 1323.94      |
> | Ours   | 0.208s | 7.203s  | 1150.66      |
>
> **Considering the one-off nature of data synthesis, the cost disparity between our method and GeoGen is negligible.**
>
> > **Q2:How can authors overcome the usability limitation of the method to be able to extend it beyond plane geometry? Detailed explanation would shed light on the future iterations.**
>
> Thank you for this question！ While NeSyGeo is currently instantiated using a domain-specific language  for plane geometry, **we posit that our methodological framework is fundamentally generic and readily transferable to other domains.**
>
> For instance, extending the framework to Analytic Geometry would entail the following steps:
> - DSL Definition: Defining a new DSL that incorporates primitives specific to the analytic space, such as "coordinates" and "functions" (e.g., defining operations like Exp_func(f_1) = (k, α, x).
> - Implementing a new Painter module to render coordinate systems and curves.
> - Adapting the Reasoner/Verifier: Since LLMs already possess pre-trained knowledge of analytic geometry, these modules can be rapidly aligned to generate CoT based on the new DSL.
>
> > **Q3:What happens if we swap the teacher models with smaller models? Does the performance drop drastically? Is there a minimum capability threshold that a model should have to be considered safe?**
>
> Thanks for this great insight! **To investigate the sensitivity of our framework to model capacity, we conducted an ablation study using smaller open-source models (Qwen3-8b and Qwen3-32b) as the Reasoner and Verifier.** We generated 600 samples for each configuration and invited 10 experts to evaluate them following the protocol outlined in Appendix C.2.
>
> | **Reasoner** | **Verifier** | **Reasoning Correctness** | **Reasoning Fluency** | **Overall Reality** | **Overall Difficulty** |
> | ------------ | ------------ | ------------------------- | --------------------- | ------------------- | ---------------------- |
> | Qwen3-32b    | Qwen3-32b    | 4.83                      | 4.76                  | 4.56                | 4.67                   |
> | Qwen3-8b     | Qwen3-8b     | 4.53                      | 4.63                  | 4.03                | 3.60                   |
> | DeepSeek-R1  | Qwen3-8b     | 4.70                      | 4.63                  | 4.56                | 3.90                   |
> | DeepSeek-R1  | Qwen3-32b    | 4.90                      | 4.90                  | 4.10                | 4.00                   |
> | Qwen3-8b     | DeepSeek-V3  | 4.90                      | 4.80                  | 4.23                | 4.30                   |
> | Qwen3-32b    | DeepSeek-V3  | 4.63                      | 4.80                  | 4.53                | 4.50                   |
> | DeepSeek-R1  | DeepSeek-V3  | 4.57                      | 4.34                  | 4.27                | 4.23                   |
>
> The results indicate that **using smaller models (e.g., 8B) does not lead to a sharp performance decline**, which we attribute to the robustness of our neuro-symbolic framework.  However, the 8B model exhibits slightly lower scores in Overall Difficulty and Overall Reality compared to others.
>
> **Therefore, while the framework remains viable with smaller models, we recommend using models at the 32B scale or larger to ensure optimal data complexity and fidelity.** We have included detailed visualizations of these comparative results in Appendix C.3.

---

> ### Author Response · Authors · 2025-11-20
> **Response to Reviewer ybzn (Part 2)**
>
> > **Q4:RE information orthogonality in 3.4, what is the optimal balance between text and image modalities?  How performance would change if we include such data in text?**
>
> Thanks for your good question! As highlighted in Figure 1, previous studies often provided redundant data or biased information heavily toward the text modality.**This tendency inadvertently creates "text shortcuts," casting doubt on the genuine contribution of visual processing in the reasoning pipeline**.
>
> To mitigate this imbalance in Geometry Problem Solving , our "Information Orthogonality" strategy **deliberately restricts all numerical values and geometric annotations to the visual modality, reserving the text solely for abstract entity descriptions and relationships.** We posit that this separation forces the model to actively synthesize information from both modalities, fostering robust cross-modal reasoning.
>
> To empirically investigate the question "What if we included redundant numerical data in the text?", we constructed a contrastive dataset,  which augments the textual descriptions with all numerical annotations present in the images.
>
> | Dataset | TD-Angle | TD-Area | TD-Length | TD-Plane | VO-Angle | VO-Area | VO-Length | VO-Plane |
> | :--- | :---: | :---: | :---: | :---: | :---: | :---: | :---: | :---: |
> | Base | 47.1 | 27.5 | 43.4 | 44.1 | 24.4 | 20.9 | 29.1 | 27.3 |
> | NeSyGeo | 49.2 | 28.6 | 44.0 | 45.5 | 29.0 | 27.5 | 34.1 | 31.8 |
> | NeSyGeo+RED | 52.8 | 27.5 | 44.5 | 45.1 | 27.5 | 25.3 | 33.0 | 30.2 |
> | R-CoT | 51.8 | 24.2 | 45.0 | 46.5 | 25.9 | 23.1 | 31.3 | 29.2 |
>
> - Text-Dominant (TD): The NeSyGeo+RED (redundant) model performs marginally better. This is expected, as the model is trained to exploit textual shortcuts without relying on visual grounding.
>
> - Vision-Only (VO) : Crucially, the model trained on our original NeSyGeo  significantly outperforms the redundancy-trained model, which confirms that our orthogonality constraint effectively prevents reliance on text shortcuts and compels the model to develop superior visual perception capabilities.
>
> ### References
>
> [1] Y. Pan et al., "Enhancing the Geometric Problem-Solving Ability of Multimodal LLMs via Symbolic-Neural Integration," ACM MM, 2025.

---

> ### Comment · Reviewer_ybzn · 2025-11-25
>
> I really appreciate authors addressing my concerns and moreover adding extra experiments. I will maintain my score and vote in favor of acceptance as I believe this is a great work.

---

### Official Review · Reviewer_XMfm · 2025-10-31

**Soundness:** 2
**Presentation:** 3
**Contribution:** 2
**Rating:** 4
**Confidence:** 3

**Summary:**

This paper presents NeSyGeo, a neural-symbolic hybrid framework, for automatically generating multimodal geometric reasoning data. Through self-developed Geo-DSL symbolic language, a question-generation pipeline of reverse search and forward verification, as well as a visual-text decoupling strategy, a 100,000-scale Caption/CoT training set and 2,668 evaluation benchmarks were constructed. 4k sample RL fine-tuning can bring a maximum 15.8% improvement on benchmarks such as MathVision, and a 4B model can outperform its equivalent 8B model.

**Strengths:**

1.For the first time, differentiable neural search was combined with strict symbolic constraints, achieving a balance between diversity and correctness.

2.Geo-DSL can fully express plane geometry with only 37 primitives, and the interpretation is unambiguous.

3.The two-stage generation of reverse search + forward verification significantly reduces hallucinations, and the quality of manual evaluation is leading.

4.The orthogonal design of visual-text information forces the model to truly read the image, alleviating the modal shortcut.

5.Data is plug-and-play, and 4k sample RL can significantly improve performance, with friendly resource utilization.

**Weaknesses:**

1. Geo-DSL only defines the basic elements of planar Euclidean geometry, lacking the symbolic primitives for coordinates, vectors, solid geometry and analytic geometry. This results in a significant disparity between the types of questions generated and those found in real exams and textbooks, especially in terms of comprehensive questions.

2.The lengths and angles are uniformly sampled within a fixed range, and the angles are forced to be multiples of 15°. This results in the frequencies of special angles (30°, 45°, 60°) being artificially increased, while common non-special angles, irrational lengths, and extreme ratios (such as 1:100) rarely appear in the actual test questions. This makes the model's robustness questionable when it truly generalizes to "irregular" shapes.

3.The values of the element selection matrix I and the action weight matrix A, the temperature τe/τa, as well as the length/angle discretization intervals were all set by the author based on experience. There was no basis for automatic learning or grid search. Once the geometric domain or difficulty distribution is changed, the parameters need to be manually adjusted again, which reduces the transferability and scalability of the method.

4.The paper emphasizes "ensuring correctness", but it has not conducted difficulty equivalence verification with symbol solvers such as AlphaGeometry and InterGPS, nor has it randomly generated questions for the symbol solvers to run and reported the solvable rate and the number of solving steps. The lack of such a comparison makes it difficult to prove that the generated questions indeed have provability and appropriate complexity.

**Questions:**

see weakness

---

> ### Author Response · Authors · 2025-11-20
> **Response to Reviewer XMfm (Part 1)**
>
> > **Q1:Geo-DSL only defines the basic elements of planar Euclidean geometry, lacking the symbolic primitives for coordinates, vectors, solid geometry and analytic geometry.**
>
> Thank you for this insight! Plane geometry solving is widely recognized as **a challenging and unsolved domain** for MLLMs and recent works have made valuable exploratory strides.
>
> NeSyGeo is founded upon a **Domain-Specific Language**, and we posit that establishing a high-fidelity, high-quality, and large-scale dataset for this specific domain **constitutes a significant contribution** in itself.
>
> Furthermore, the NeSyGeo framework is **highly extensible**. Our underlying methodology (DSL Definition $\to$ Generator $\to$ Solver $\to$ Painter) can be readily adapted to other mathematical domains.
>
> For instance, extending the framework to generate Analytic Geometry problems would only require:
> - Defining a new DSL that incorporates primitives such as "coordinates" and "functions."
> - Implementing a new Painter module to render coordinate systems and curves.
> - Adapting the Reasoner/Verifier: Since LLMs already possess pre-trained knowledge of analytic geometry, these modules can be rapidly aligned to generate CoT based on the new DSL.
>
> > **Q2: Extreme ratios (such as 1:100) rarely appear in the actual test questions. This makes the model's robustness questionable when it truly generalizes to "irregular" shapes.**
>
> Thanks for your question! Our design choices were driven by the following considerations:
>
> - Alignment with Real-World Distributions: We considered the distribution of problems in **existing real-world benchmarks** (e.g., MathVerse, GeoQA). These exhibit a strong bias toward special angles and standard ratios, which serve as **the foundation for trigonometric functions and key geometric theorems**.
>
> - Task Integrity : We aimed to prevent the task from devolving into **pure numerical approximation**, as plane geometry solving task is fundamentally characterized by deductive reasoning based on given conditions and geometric theorems. Using fully random angles often results in **analytically unsolvable problems** (e.g., requiring $\sin(17.3^{\circ})$), which would distract from the core objective of **evaluating visual perception and geometric reasoning**.
>
>  We also considered the necessity of a flexible framework. While we prioritized fundamental problems for this initial release, our hyperparameters (e.g., angle weights) can be easily adjusted. We posit that mastering these "regular" problems is a necessary prerequisite before models can effectively generalize to irregular cases requiring numerical approximation.
>
> > **Q3:  There was no basis for automatic learning or grid search. Once the geometric domain or difficulty distribution is changed, the parameters need to be manually adjusted again, which reduces the transferability and scalability of the method.**
>
> Thank you for this insightful suggestion!
>
> Automatic learning or grid search is relatively rare in the context of LLM/MLLM data synthesis, primarily due to **the prohibitive computational costs** associated with training and evaluation iterations.
>
> Our current configuration ensures **a balanced distribution of geometric topics** (as illustrated in Figure 6) and **maintains controllability over the overall difficulty**.
>
>  However, our framework remains flexible; for dataset expansion or customization, the following parameters can be adjusted:
>
> - Element Weights ($\mathbf{I}_j$): Regulates the global prevalence of specific geometric primitives.
> - Action Transition Weights ($\mathbf{A}_{k,j}$): The conditional probabilities of constructive steps, thereby guiding the logical flow and complexity of the generated diagrams.
> - Selection Temperatures ($\tau_e, \tau_a$): The entropy of the sampling distributions, balancing the trade-off between high diversity and standard, stereotypical constructions
>
> Nonetheless, we agree that the automated process suggested is highly promising.
>
> Our framework is  compatible with a "Generator-Evaluator" feedback loop. In such a system, a Meta-Controller could automatically tune the generator parameters ($\mathbf{I}, \mathbf{A}, \tau$) using reward signals derived from downstream MLLM performance metrics. This would allow the data synthesis process to self-optimize toward specific training objectives. We intend to actively explore this direction in future work.

---

> ### Author Response · Authors · 2025-11-20
> **Response to Reviewer XMfm (Part 2)**
>
> > **Q4:.The paper emphasizes "ensuring correctness", but it has not conducted difficulty equivalence verification with symbol solvers.**
>
> We appreciate your feedback! In our framework, the  **Verifier  guarantees data correctness**. This reliability is further corroborated by our human expert evaluation results **shown in Figure 4**.
>
> To provide the external symbolic validation suggested by the reviewer, we utilized open-source symbolic solvers, specifically InterGPS [1] and Pi-GPS [2]. Note that **AlphaGeometry was excluded as it does not support numerical metric calculations**. We sampled 20 questions from our dataset and **manually translated the conditions and queries** into the solvers' symbolic formats.
>
> | Method       | Pass Rate | Average Steps |
> | ---------------- | ------------- | ----------------- |
> | Inter-GPS        | 20.00%        | 31.50             |
> | Pi-GPS           | 30.00%        | 5.33              |
> | Human  | 100.00%       | 4.75              |
>
> A detailed case analysis reveals that failures are primarily attributable to **limitations within the solvers themselves**: 20% due to missing theorems, 20% due to target parsing failures, and 40% due to internal solver errors. Furthermore, the high average step count required for symbolic solutions **validates the complexity and quality of our data**. We have provided detailed case studies in Appendix C.4 in our revised scripts.
>
>
> ### References
>
> [1] P. Lu et al., "Inter-GPS: Interpretable Geometry Problem Solving with Formal Language and Symbolic Reasoning," ACL, 2021.
>
> [2] J. Zhao et al, "Pi-GPS: Enhancing Geometry Problem Solving by Unleashing the Power of Diagrammatic Information," ICCV,2025.

---

### Official Review · Reviewer_VMe1 · 2025-11-01

**Soundness:** 3
**Presentation:** 3
**Contribution:** 3
**Rating:** 4
**Confidence:** 4

**Summary:**

This paper propose NeSyGeo, a neural-symbolic framework for generating plane geometry problems. The NeSyGeo use DSL to describe the elements in the plane geometry, which provides an explicit basic problem information for the data generation process. The problem generation algorithm use both reasoner and verifier to ensure the method output the correct problem with CoT solutions, which ensure the usability of the generated problems. Based using the generated dataset to train the model, smaller MLLM get higher accuracy than larger models.

**Strengths:**

1.	The definition of the Geo-DSL is reasonable, which covered primitive entities, metric attributes and topological relations.
2.	The problem generation process is reliable, which has backward search and forward verification and finally get the general language description of the CoT process. With the diagram painter to get the entire generated problem text, diagram and solution.
3.	Based on the experiments, the generated dataset help model to deeper rely on diagram to solve the problem, different from the previous methods that mainly based on problem text, like shown in Figure 1.
4.	The author contributed to dataset, NeSyGeo-CoT and NeSyGeo-Caption, from the experiments, the proposed dataset and method are effective, and the tuned models based on these datasets achieved increased performance on benchmarks, the 4B MLLM is compatible with largers MLLMs.

**Weaknesses:**

1.	As a plane geometry problem generation work, the part of how to translate the DSL into a diagram is missing, how to get the location or coordinates of the points in the diagram? Meanwhile, is it possible to generate diagrams with only lines and no closed geometry shapes in the diagram, like a problem with two parallel lines and another line across them to ask about alternate angles.
2.	The experiments for a dataset generation work are essential to show the effectiveness of the proposed dataset, and the current experiments are not enough, especially for the SFT. It is only comparing with the base MLLMs. Like comparing to R-CoT, from their work the result of SFT on Qwen2.5-VL-7B get 79.2% on GeoQA, which has a huge gap between 71.8 in NeSyGeo, the author needs further experiments on this kind of comparsion.
3.	Why is Geometry3K missing in the experiments? It is also an important foundation dataset for the research of PGP area. Despite the current experiments showing that with the proposed new dataset, MLLMs achieved increased performance, I am wondering about the effectiveness of this method on Geometry3K, as the PGP style of GeoQA and Geometry3K are different. There are some open-source works like EasyR1 provides the GRPO training for Geometry3K, it is better to provide a compression with directly use GRPO to traing MLLM and use the proposed dataset to RL train the MLLM. This experiment will be more convincing to the work.
4.	Is it possible to have a further ablation experiment on using different ratios of the proposed dataset to train the model and analyze the model performance, like 20%, 50%, 80%, to show whether the increase in training samples leads to a linear performance increase.
5.	Some minor revision points. The dataset name is nor accurate, like for G-LLaVA, the corresponding dataset is Geo170K. For R-CoT, the new version is now named TR-CoT, and the dataset name is TR-GeoMM. Just suggest refining them. Some tables are too large.
6.	Finally, my concerns are mainly about the experiments， if the above concerns are tackled by the author, I will lean to accept the paper.

**Questions:**

1.	When draw the visual diagram with annotations, if the diagram is complicated, how do you avoid the annotation is overlapping to the diagram lines.
2.	What is the upper limit of the method, is it possible to generate complicated problems and diagrams, like diagrams in AlphaGeometry, the IMO level.

---

> ### Author Response · Authors · 2025-11-20
> **Response to Reviewer VMe1 (Part 1)**
>
> We sincerely appreciate your insightful reviews and constructive advice. We have provided detailed responses to your comment and updated the relevant content in the revised manuscript, hoping to address your concerns.
> > **Q1:How to get the location or coordinates of the points in the diagram? Is it possible to generate diagrams with only lines and no closed geometry shapes in the diagram?**
>
> Thanks for your question! To determine point locations and render diagrams, Painter  mainly focuses on following pipelines:
> - **Deterministic Coordinate Instantiation**: The engine initiates by anchoring a primary entity to the origin and treats DSL statements as algebraic constraints. It sequentially propagates the constraints by using trigonometric functions and affine transformations to compute the exact spatial coordinates of all dependent elements, resulting in **a precise Geometric Primitive Dictionary** that encodes the precise coordinates, attributes, and topology of all geometric entities。
> - **Robust Visual Instantiation**: To ensure fidelity and diversity, the engine performs viewport normalization to shift the geometric constellation into the positive quadrant. Additionally, stochastic strategies are applied to sample global scales, translations, and color palettes to **mitigate overfitting to specific spatial biases**.
>
> We have provided a comprehensive description of these implementation details in the revised 3.3 and Appendix A.3.
>
> As for the diagram with only lines and no closed geometry shapes, it can be constructed using **only Geo-DSL statements related to lines and angles**. We have included a generated example named 'Alternate Interior Angles Theorem' in Figure 19 of the revised manuscript for your reference.
>
> > **Q2:Results on GeoQA benchmark compared with R-CoT.**
>
> Thanks for pointing it out!
>
> We thank the reviewer for bringing the latest version of R-CoT(TR-CoT) [1] to our attention. We have carefully reviewed their work and offer the following clarifications.
>
> 1. Dataset Composition
>
> The 79.2% SFT accuracy reported in TR-CoT is achieved by training on **a joint dataset combining Geo170K (117k real-world samples)** and their synthetic TR-GeoMM/Sup data. In contrast, the 71.8% result reported in our original submission was achieved using exclusively our synthetic NeSyGeo data, without any human-annotated samples.
>
> 2. Evaluation Protocols
>
> We note that performance discrepancies often arise from differing evaluation methodologies. For instance, while TR-CoT reports 74.5% for the Qwen2.5-VL-7B base model on GeoQA, other works[2]  report only 64.3%. Our evaluation  follows  "llm-as-a-judge" protocol using GPT-4o to ensure objectivity.
>
> To provide a direct and fair comparison, we conducted a controlled experiment using the Geo170K instruction-tuning split (117k samples) as the baseline. We supplemented this baseline with an equal volume (30k) of our NeSyGeo data versus the TR-CoT  data (TR-GeoMM). The results on GeoQA are presented below:
>   | Training Datasets | Accuracy (%) |
>   | :-- | :-- |
>   | Qwen2.5-VL-7B (Base) | 69.4 |
>   | Geo170K | 72.3 |
>   | Geo170K + TR-GeoMM | 76.5 |
>   | Geo170K + NeSyGeo | **77.2** |
> We have also formally cited the TR-CoT  to ensure a comprehensive comparison.
>
> > **Q3: It is better to provide a compression with directly use GRPO to traing MLLM and use the proposed dataset to RL train the MLLM.**
>
> Thank you for this observation! We excluded Geometry3K from the training comparison because it primarily relies on manual annotation, lacking the scalability inherent to automated synthesis.
>
> However, we supplemented our experiments by evaluating the model trained via GRPO (on 4,000 samples) on the Geometry3K test set.
>
> As shown in the results below, the model demonstrates consistent capability improvements across diverse GPS styles. This further validates the high quality and generalization potential of our dataset.
>
> | Methods     | Base | NeSyGeo | TR-CoT | MAVIS |
> | ------------ | ---- | ------- | ------ | ----- |
> | **Accuracy(%)** | 34.1 | 40.1   | 37.4  | 38.6 |

---

> ### Author Response · Authors · 2025-11-20
> **Response to Reviewer VMe1 (Part 2)**
>
> > **Q4: Is it possible to have a further ablation experiment on using different ratios of the proposed dataset to train the model and analyze the model performance, like 20%, 50%, 80%?**
>
> Thank you for the suggestion! To investigate the impact of data scale,  we randomly sample 30%, 50% and 80% and 100% instruction tuning data combining Geo170K (117k real-world samples) to train Qwen2.5-VL-7B-Instruct in SFT and evaluated the results on the GeoQA benchmark.
>
> As shown in the table below, the accuracy of Qwen2.5-VL-7B **consistently improves as the data volume increases**. This scaling behavior validates the **high quality and scalability** of our generation framework, demonstrating the dataset's significant potential for further enhancing mathematical reasoning capabilities.
>
> | Trainset      | Base  | Geo170k | +30% Ours | +50% Ours | +80% Ours | +100% Ours |
> | ------------ | ----- | ------- | --------- | --------- | --------- | ---------- |
> | **Accuracy** | 69.4% | 72.3%   | 72.8%     | 74.3%     | 76.0%     | 77.2%      |
>
> We also report the accuracy trends across multiple datasets during the GRPO training of Qwen2.5-VL-3B-Instruct with 4,000 samples of NeSyGeo-CoT, which further validates the quality and stability of our dataset.
>
> | **Dataset**                          | **Metric**     | **Base** | **0.2 epoch** | **0.6 epoch** | **1.0 epoch** | **1.4 epoch** | **2 epoch** |
> | ------------------------------------ | -------------- | -------- | ------------- | ------------- | ------------- | ------------- | ----------- |
> | **MathVerse** **(Vision Only)**      | Angle          | 24.4     | 25.9          | 28.5          | 28.0          | 28.0          | 29.0        |
> |                                      | Area           | 20.9     | 22.0          | 20.9          | 25.3          | 26.4          | 27.5        |
> |                                      | Length         | 29.1     | 31.3          | 30.2          | 33.0          | 33.5          | 34.1        |
> |                                      | Plane Geometry | 27.3     | 28.0          | 29.0          | 29.8          | 30.8          | 31.8        |
> | **MathVerse** **(Vision Dominant)**  | Angle          | 33.2     | 38.4          | 36.7          | 36.8          | 45.6          | 45.1        |
> |                                      | Area           | 25.3     | 24.2          | 23.1          | 26.4          | 25.3          | 26.4        |
> |                                      | Length         | 33.7     | 36.2          | 36.2          | 35.2          | 35.7          | 36.2        |
> |                                      | Plane Geometry | 32.9     | 36.1          | 34.7          | 33.7          | 37.3          | 38.2        |
> | **MathVerse** **(Vision Intensive)** | Angle          | 36.8     | 36.8          | 37.3          | 38.9          | 45.1          | 45.6        |
> |                                      | Area           | 22.0     | 19.8          | 23.1          | 24.2          | 23.1          | 20.9        |
> |                                      | Length         | 33.0     | 33.5          | 34.1          | 40.1          | 36.3          | 37.9        |
> |                                      | Plane Geometry | 31.8     | 35.3          | 35.8          | 37.1          | 39.4          | 40.2        |
>
> > **Q5: Some minor revision points.**
>
>  Thanks for this careful observation!  We have updated the dataset names to their accurate versions and resized the tables in the revised manuscript.
>
> > **Q6: When draw the visual diagram with annotations, if the diagram is complicated, how do you avoid the annotation is overlapping to the diagram lines?**
>
> Thank you for the question!  We implemented specific programmatic measures to minimize annotation overlap:
>
> - **Positional Constraints**: Annotations follow deterministic default placement strategies rather than random assignment:
>   - Length: Labels are placed at the midpoint of the corresponding line segment, offset outward along the normal vector by a fixed distance.
>   - Angle: Labels are positioned on the angle bisector at a fixed radius from the vertex.
> - **Overlap Detection**: To address potential conflicts, we employ iterative collision detection during rendering. We compute the bbox of the label and check for intersections with key geometric elements. If an overlap is detected, we apply fine-tuning adjustments (switching to "inward" labeling or applying slight multidirectional offsets) until the collision check passes.
>
> As validated by our human preference experiments in Figure 4 and other case analysis in appendix, these measures effectively ensure **the legibility of annotations** and **the high quality of images** within the NeSyGeo dataset.

---

> ### Author Response · Authors · 2025-11-20
> **Response to Reviewer VMe1 (Part 3)**
>
> > **Q7: What is the upper limit of the method, is it possible to generate complicated problems and diagrams, like diagrams in AlphaGeometry, the IMO level?**
>
> Thank you for the question. We explain the complexity of our data in two parts:
>
> 1. Diagram Complexity:
>
> Our method is capable of generating IMO-level complex imagery. Due to the iterative nature of our Generator, we can synthesize geometric diagrams of **arbitrary complexity by increasing the number of generation steps** in the DSL sequence. This includes the sophisticated structures seen in AlphaGeometry. To demonstrate this, we randomly selected geometry diagrams from historical IMO competitions and synthesized them using our GEO-DSL and Painter; these results are visualized in Figure 19 of the revised manuscript.
>
> 2. Problem Complexity:
>
> The problems of NeSyGeo are generated by the Reasoner and Verifier modules, which currently rely on LLMs. Thanks to our step-wise DFS search and sampling strategies, our method is robust for problems at the high school levels.
>
> However, we acknowledge a limitation regarding higher-difficulty problems, such as those at the IMO level. These problems typically require highly creative auxiliary line constructions. Reliably generating and verifying such creative leaps is currently beyond the capabilities of LLMs within our framework.
>
> ### References
>
> [1] L. Deng et al., "Theorem-Validated Reverse Chain-of-Thought Problem Generation for Geometric Reasoning," EMNLP, 2025.
>
> [2] Y. Pan et al., "Enhancing the Geometric Problem-Solving Ability of Multimodal LLMs via Symbolic-Neural Integration," ACM MM, 2025.

---

### Author Response · Authors · 2025-11-26
**Revision Summary**

We sincerely thank the reviewers for their constructive feedback and thorough evaluations. Please refer to our **individual responses** for point-by-point replies to specific concerns. Guided by your suggestions, we have updated the manuscript, with all significant revisions **highlighted in blue**.

**Sec. 3.4**: Added a brief clarification on **Painter's two-stage process involving Deterministic Coordinate Instantiation and Robust Visual Instantiation.**

**Sec. 4.2**: Updated dataset nomenclature to ensure **more accurate descriptions**.

**Sec. 4.2 -- Performance**: Added **SFT scaling experiments** (Table 4) to further verify the scalability of our dataset.

**Appendix A.3**: Provided a detailed elaboration on **Painter’s two-stage workflow and its specific advantages**.

**Appendix C.3**: Included **ablation studies on model selection for the Reasoner and Verifier** to investigate performance boundaries.

**Appendix C.4**: Added experiments **using a symbolic solver to verify the correctness and difficulty of our dataset.**

**Appendix E.2**: Analyzed the data quality of symbolic solver-based methods to **highlight their specific limitations.**

**Appendix H.1**: Added examples of **geometry figures across varying difficulty levels synthesized via Geo-DSL** to demonstrate the completeness of our definitions.

We believe these revisions improve clarity and faithfully reflect the scope of our study. **Thank you again for your insightful feedback and constructive suggestions**.

---

### Author Response · Authors · 2025-12-02
**Summary of Rebuttal**

Dear  AC,

Thank you for taking over the handling of this submission. We would like to briefly summarize the key points of our rebuttal.

Through additional experiments and thorough analyses, we have effectively addressed concerns like  **data scaling, method generalizability, correctness assurance, and novelty distinctions**. We summarize the key strengths of NeSyGeo highlighted by the reviewers below:

- **Novel Neuro-Symbolic Paradigm:** The Reasoner-Verifier architecture effectively combines differentiable neural search with strict symbolic constraints, achieving an optimal balance between generation diversity and logical correctness. (Reviewers VMe1, ybzn, XMfm)
- **Solves Modal Redundancy:** The "Information Orthogonality" strategy forces models to rely on visual perception, effectively mitigating the "text shortcut" issue prevalent in existing datasets. (Reviewers VMe1, ybzn, XMfm)
- **High Efficiency & SOTA Performance:** The framework is highly efficient, enabling a 4B model to outperform 8B baselines using only 4k RL samples, and showing consistent gains across diverse benchmarks. (Reviewers VMe1, ybzn, 4nkp)
- **High-Quality & Hallucination-Free:** The defined Geo-DSL is unambiguous and comprehensive, ensuring that generated CoT paths are logically sound yet linguistically diverse, significantly reducing hallucinations compared to pure LLM approaches. (Reviewers VMe1, XMfm)
- **Comprehensive Experimental Validation:** The paper provides extensive validation proving the dataset's reliability and plug-and-play utility. (Reviewers 4nkp, ybzn, VMe1)

We are pleased to receive final positive feedback from **Reviewer ybzn**, stating they will **"vote in favor of acceptance"** following our additional experiments. However, due to unforeseen circumstances, we have not yet received final confirmations on some of our other responses:

- **Reviewer VMe1:** The reviewer stated they would **"lean to accept the paper"** if their experimental concerns were tackled. We have explicitly addressed this by providing comprehensive experimental data, including data scaling analysis and performance validation on the Geometry3K benchmark.
- **Reviewer XMfm:** We addressed concerns regarding **Geo-DSL extensibility** (e.g., to analytic geometry) and **validated the difficulty of our data** through comparative experiments with symbolic solvers such as InterGPS. **We also supplemented relevant analysis on the limitations of pure symbolic solver methods in our revised scripts.**
- **Reviewer 4nkp:** We provided further clarification on **the distinct advantages of NeSyGeo over AlphaGeometry and TrustGeoGen**, specifically highlighting our unique capability for numerical VQA generation and unbounded topological diversity.

Overall, NeSyGeo contributes a high-quality and large-scale dataset for multimodal geometric reasoning, which guarantees both data correctness and diversity via a controllable neuro-symbolic synthesis framework. We have substantively addressed the reviewers' concerns and hope the AC will consider the resulting consensus on the framework's effectiveness and our detailed clarifications in the final evaluation.

We sincerely appreciate your time and efforts throughout the review process.

Best regards.

---

### Meta-Review · Area_Chair_PvyR · 2026-01-06

**Summary:**

The paper proposes NeSyGeo, a neuro-symbolic framework designed to generate multimodal geometric reasoning data. The method utilizes a Domain-Specific Language (Geo-DSL) combined with a Reasoner-Verifier pipeline and a Painter module to synthesize problems and solutions. While the reviewers acknowledged the extensive experiments and the effectiveness of the generated dataset in improving MLLM performance on specific benchmarks, the decision is to Reject. The primary reasons for this decision are the limited scope of the framework (strictly plane geometry) and persistent concerns regarding the lack of significant methodological novelty compared to existing neuro-symbolic systems like AlphaGeometry or TrustGeoGen.

**Reviewer Concerns:**

**Addressed Concerns:**
*   **Reviewer VMe1** and **Reviewer ybzn**'s questions regarding the technical details of the "Painter" module and the coordinate instantiation process were clarified in the revision.
*   **Reviewer VMe1**'s request for additional data scaling experiments and comparisons on the Geometry3K benchmark was addressed with new experimental results.
*   **Reviewer XMfm**'s concern regarding the lack of comparison with symbolic solvers was mitigated by the inclusion of experiments involving InterGPS and Pi-GPS.
*   **Reviewer ybzn**'s concern about inference latency was addressed through efficiency comparisons with GeoGen.

**Outstanding Concerns:**
*   **Reviewer 4nkp** remains unconvinced about the novelty of the framework, noting that the advantages over established methods like AlphaGeometry and TrustGeoGen are not sufficiently distinct.
*   **Reviewer 4nkp** highlights that the reliance on stronger teacher models (DeepSeek) for reasoning limits the difficulty ceiling and renders the approach a form of distillation rather than a fundamental generation breakthrough.
*   **Reviewer XMfm** pointed out the limitation of the Geo-DSL being restricted to plane geometry (lacking analytic or solid geometry primitives), which significantly limits the framework's generalizability.
*   **Reviewer XMfm** noted that the artificial distribution of geometric attributes (e.g., angles as multiples of 15) may not reflect real-world problem diversity and could hamper robustness.

**Reviewer Scores:**

*   **Reviewer VMe1: 6**
    The reviewer indicated a willingness to lean towards acceptance after the authors provided the requested experimental comparisons and implementation details.

*   **Reviewer XMfm: 4**
    The score would likely maintain as the limitation regarding the specific scope of plane geometry remains, although the authors addressed the comparison with symbolic solvers.

*   **Reviewer ybzn: 6**
    The reviewer explicitly stated they would maintain their score, appreciating the additional experiments but noting the method's complexity and domain limitations.

*   **Reviewer 4nkp: 4**
    The reviewer explicitly maintained their original rating, asserting that the rebuttal did not resolve the fundamental concerns regarding novelty and the distinction from existing symbolic-neural approaches.

---

### Decision · Program_Chairs · 2026-01-26

Reject